# A Coefficient Makes SVRG Effective

**Yida Yin**[1]    **Zhiqiu Xu**[2]    **Zhiyuan Li**[3]    **Trevor Darrell**[1]    **Zhuang Liu**[4]

[1]UC Berkeley    [2]University of Pennsylvania    [3]TTIC    [4]Meta AI Research

## Abstract

Stochastic Variance Reduced Gradient (SVRG), introduced by Johnson & Zhang (2013), is a theoretically compelling optimization method. However, as Defazio & Bottou (2019) highlight, its effectiveness in deep learning is yet to be proven. In this work, we demonstrate the potential of SVRG in optimizing real-world neural networks. Our empirical analysis finds that, for deeper neural networks, the strength of the variance reduction term in SVRG should be smaller and decrease as training progresses. Inspired by this, we introduce a multiplicative coefficient $\alpha$ to control the strength and adjust it through a linear decay schedule. We name our method $\alpha$-SVRG. Our results show $\alpha$-SVRG better optimizes models, consistently reducing training loss compared to the baseline and standard SVRG across various model architectures and multiple image classification datasets. We hope our findings encourage further exploration into variance reduction techniques in deep learning. Code is available at `github.com/davidyyd/alpha-SVRG`.

## 1 Introduction

A decade ago, Johnson & Zhang (2013) proposed a simple approach for reducing gradient variance in SGD—Stochastic Variance Reduced Gradient (SVRG). SVRG keeps a snapshot model and uses it to form a variance reduction term to adjust the gradient of the current model. This variance reduction term is the difference between the snapshot's stochastic gradient and its full gradient on the whole dataset. Utilizing this term, SVRG can reduce gradient variance of SGD and accelerate it to almost as fast as the full-batch gradient descent in strongly convex settings.

Over the years, numerous SVRG variants have emerged. Some focus on further accelerating convergence in convex settings (Xiao & Zhang, 2014; Lin et al., 2015; Defazio, 2016), while others are tailored for non-convex scenarios (Allen-Zhu & Hazan, 2016; Reddi et al., 2016; Lei et al., 2017; Fang et al., 2018). SVRG and its variants have shown effectiveness in optimizing simple machine learning models like logistic regression and ridge regression (Allen-Zhu, 2017; Lei et al., 2017).

Despite the theoretical value of SVRG and its subsequent works, they have seen limited practical success in training neural networks. Most SVRG research in non-convex settings is restricted to modest experiments: training basic models like Multi-Layer Perceptrons (MLP) or simple CNNs on small datasets like MNIST and CIFAR-10. These studies usually exclude evaluations on more capable and deeper networks. More recently, Defazio & Bottou (2019) have exploited several variance reduction methods, including SVRG, to deep vision models. They found that SVRG fails to reduce gradient variance for deep neural networks because the model updates so quickly on the loss surface that the snapshot model becomes outdated and ineffective at variance reduction.

In this work, we show that adding a multiplicative coefficient to SVRG's variance reduction term can make it effective for deep neural networks. Our exploration is motivated by an intriguing observation: SVRG can only reduce gradient variance in the initial training stages but *increases* it later. To tackle this problem, we mathematically derive the optimal coefficient for the variance reduction term to minimize the gradient variance. Our empirical analysis then leads to two key observations about this optimal coefficient: (1) as the depth of the model increases, the optimal coefficient becomes smaller; (2) as training advances, the optimal coefficient decreases, dropping well below the default coefficient of 1 in SVRG. These findings help explain why a constant coefficient of 1 in SVRG, while initially effective, eventually fails to reduce gradient variance.

Based on these observations, we introduce a linearly decaying coefficient $\alpha$ to control the strength of the variance reduction term in SVRG. We call our method $\alpha$-SVRG and illustrate it in Figure 1.

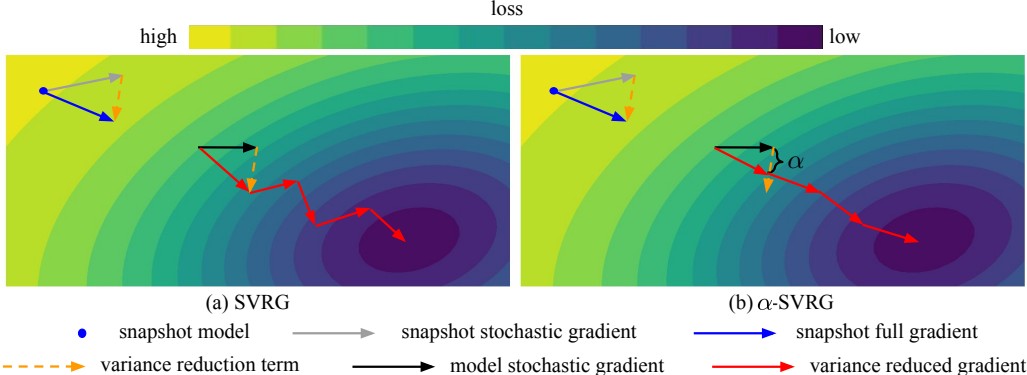

Figure 1: **SVRG vs. $\alpha$-SVRG.** Both SVRG (left) and $\alpha$-SVRG (right) use the difference between snapshot stochastic gradient (gray) and snapshot full gradient (blue) to form a variance reduction term (orange), which modifies model stochastic gradient (black) into variance reduced gradient (red). But $\alpha$-SVRG employs a coefficient $\alpha$ to modulate the strength of the variance reduction term. With this coefficient, $\alpha$-SVRG reduces the gradient variance and results in faster convergence.

$\alpha$-SVRG decreases gradient variance stably across both early and late training and helps optimize the model better. We evaluate $\alpha$-SVRG on a range of architectures and image classification datasets. $\alpha$-SVRG achieves a lower training loss than the baseline and the standard SVRG. Our results highlight the value of SVRG in deep learning. We hope our work can offer insights about SVRG and stimulate more research in variance reduction approaches for optimization in neural networks.

## 2 MOTIVATION: SVRG MAY NOT ALWAYS REDUCE VARIANCE

**SVRG formulation.** We first introduce the basic formulation of SVRG. We adopt the following notation: $t$ is the iteration index, $\boldsymbol{\theta}^t$ represents the current model parameters, and $\nabla f_i(\cdot)$ denotes the gradient of loss function $f$ for the $i$-th mini-batch. In SVRG's original work (Johnson & Zhang, 2013), this corresponds to the $i$-th data point. When the subscript $i$ is omitted, $\nabla f(\cdot)$ represents the full gradient across the entire dataset. A key concept in SVRG is the snapshot model, represented as $\boldsymbol{\theta}^{\text{past}}$. It is a snapshot of the model at a previous iteration before $t$. We store its full gradient $\nabla f(\boldsymbol{\theta}^{\text{past}})$. This snapshot is taken periodically. SVRG defines the variance reduced gradient $\boldsymbol{g}_i^t$, as follows:

$$\boldsymbol{g}_i^t = \nabla f_i(\boldsymbol{\theta}^t) - \underbrace{\left(\nabla f_i(\boldsymbol{\theta}^{\text{past}}) - \nabla f(\boldsymbol{\theta}^{\text{past}})\right)}_{\text{variance reduction term}}. \tag{1}$$

Intuitively, SVRG uses the difference between the mini-batch gradient and full gradient of a past model to modify the current mini-batch gradient. This could make $\boldsymbol{g}_i^t$ better aligned with the current full gradient $\nabla f(\boldsymbol{\theta}^t)$ and thus stabilize each update.

SVRG was initially introduced in the context of vanilla SGD. Recent work (Dubois-Taine et al., 2021; Wang & Klabjan, 2022) has integrated SVRG into alternative base optimizers. Following them, we input the variance reduced gradient $\boldsymbol{g}_i^t$ into the base optimizer and ensure a fair comparison by using the same base optimizer for SVRG and the baseline. We also follow the practice in Defazio & Bottou (2019), taking snapshot for SVRG once per training epoch.

**Gradient variance.** Our goal is to assess SVRG's effectiveness in reducing gradient variance. To this end, we gather $N$ mini-batch gradients, denoted as $\{\boldsymbol{g}_i^t | i \in \{1, \cdots, N\}\}$, by performing back-

| name | formula | description |
|---|---|---|
| metric 1* | $\frac{2}{N(N-1)} \sum_{i \neq j} \frac{1}{2}\left(1 - \frac{\langle \boldsymbol{g}_i^t, \boldsymbol{g}_j^t \rangle}{\|\boldsymbol{g}_i^t\|_2 \|\boldsymbol{g}_j^t\|_2}\right)$ | the directional variance of the gradients |
| metric 2† | $\sum_{k=1}^{d} \text{Var}(g_{i,k}^t)$ | the variance of gradients across each component |
| metric 3‡ | $\lambda_{max}\left(\frac{1}{N} \sum_{i=1}^{N}(\boldsymbol{g}_i^t - \boldsymbol{g}^t)(\boldsymbol{g}_i^t - \boldsymbol{g}^t)^T\right)$ | the magnitude of the most significant variation |

Table 1: **Metrics.** $\boldsymbol{g}^t$ is the mean of the mini-batch gradients $\boldsymbol{g}_i^t$. $k$ indexes the $k$-th component of gradient $g_{i,k}^t$. References: * Liu et al. (2023), † Defazio & Bottou (2019), ‡ Jastrzebski et al. (2020)

propagation on checkpoints of the model at the iteration $t$ with randomly selected $N$ mini-batches. For SVRG, each of these gradients is modified based on Equation 1. To present a comprehensive view, we employ three metrics from prior studies to quantify gradient variance in Table 1.

In this part, we compare our three metrics. Metric 1 calculates the cosine distance between pairwise mini-batch gradients, therefore only capturing variance in gradient directions rather than gradient magnitudes. This is very important for scale-invariant optimizers, such as Adagrad (Lydia & Francis, 2019) and Adam (Kingma & Ba, 2015). In contrast, metric 2 focuses on both gradient directions and magnitudes by summing the variance of each component of gradients. This metric has been the standard tool to measure gradient variance in various optimization literature (Allen-Zhu & Hazan, 2016; Defazio & Bottou, 2019). Metric 3 considers the largest eigenvalue in gradient covariance matrix, characterizing the most dominant part in gradient variance. We also average each gradient variance metric across three runs, with shaded regions in figures representing the standard deviation.

**SVRG's effect on gradient variance.** To understand how SVRG affects training, we examine two simple models: a linear layer (Logistic Regression) and a 4-layer Multi-Layer Perceptron (MLP-4). We train them over 30 epochs on CIFAR-10. We compare SVRG to a baseline using only SGD.

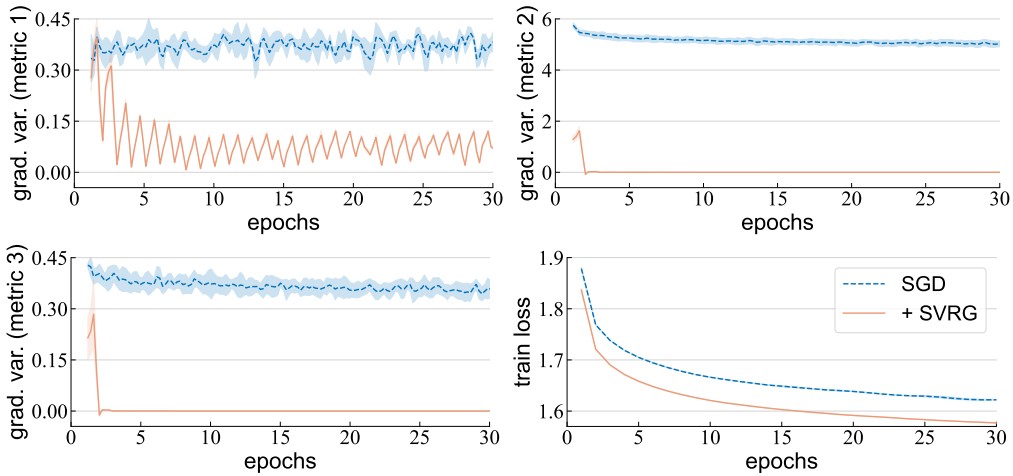

Figure 2: **SVRG on Logistic Regression.** SVRG effectively reduces the gradient variance for Logistic Regression, leading to a lower training loss than the baseline.

We plot Logistic Regression's gradient variance (top two and bottom left) and training loss (bottom right) in Figure 2. For Logistic Regression, SVRG can reduce the gradient variance throughout the entire training process and achieve a lower training loss than the baseline.

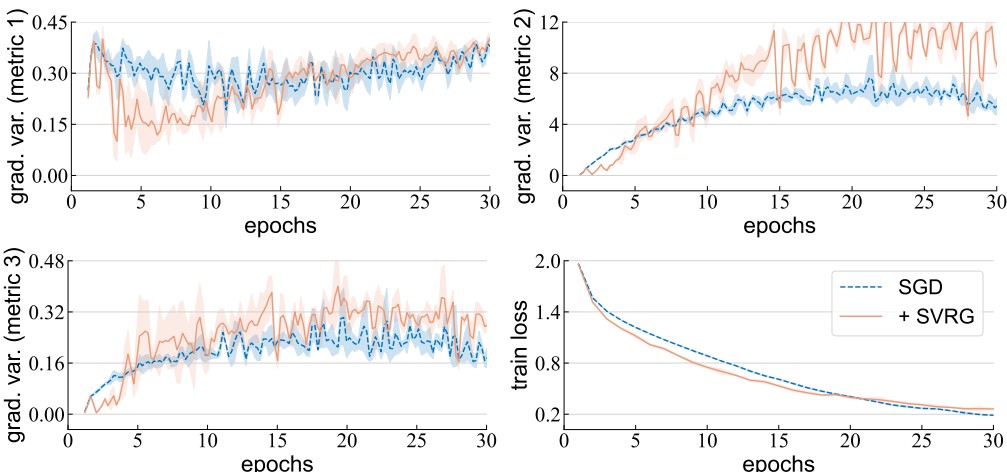

Figure 3: **SVRG on MLP-4.** In the first few epochs, SVRG reduces the gradient variance for MLP-4, but afterward, SVRG increases it, well above the baseline. As a result, SVRG exhibits a higher training loss than the baseline at the end of training.

In contrast, for MLP-4, SVRG may not always reduce gradient variance. As shown in Figure 3, SVRG can only decrease the gradient variance for the first five epochs but then increases it. Consequently, SVRG has a larger final training loss than the baseline. This indicates that the increase in gradient variance caused by SVRG hinders the convergence of MLP-4's training loss.

This surprising empirical observation in a slightly deeper model leads us to question whether SVRG may alter the gradient too excessively at certain phases of training. Can we mitigate this adverse effect? We explore these questions starting from a theoretical framework in the next section.

## 3 A CLOSER LOOK AT CONTROL VARIATES IN SVRG

Control variates (Lavenberg et al., 1977) is a technique initially developed in Monte Carlo methods to reduce variance. We aim to estimate the expected value of a random variable X. The variance of this estimate usually depends on $\mathrm{Var}(X)$. To form a less variate estimate $X^*$, we can use a control variate Y that correlates with X and a coefficient $\alpha$ to regulate the influence of Y and $\mathbb{E}[Y]$ :

$$X^* = X - \alpha(Y - \mathbb{E}[Y]). \tag{2}$$

This estimate remains unbiased for any value of $\alpha$. The coefficient that minimizes the variance of the estimate can be derived as:

$$\alpha^* = \frac{\mathrm{Cov}(X, Y)}{\mathrm{Var}(Y)} = \rho(X, Y)\frac{\sigma(X)}{\sigma(Y)}, \tag{3}$$

where $\rho(X, Y)$ represents the correlation coefficient between X and Y; $\sigma(\cdot)$ denotes the standard deviation. The derivation is detailed in Appendix A. The minimized variance becomes $\mathrm{Var}(X^*) = (1 - \rho(X, Y)^2)\mathrm{Var}(X)$. The higher the correlation is, the lower the variance of the estimate is.

Note that SVRG uses control variates to reduce variance in each component of the gradient. This variance reduction occurs at each iteration $t$. Take a closer look at Equation 1 and 2: the model stochastic gradient $f_i(\boldsymbol{\theta}^t)$ is the random variable X; the snapshot stochastic gradient $f_i(\boldsymbol{\theta}^{\mathrm{past}})$ is the control variate Y; and the snapshot full gradient $f(\boldsymbol{\theta}^{\mathrm{past}})$ is the expectation $\mathbb{E}[Y]$.

A key difference between SVRG and control variates is that SVRG omits the coefficient $\alpha$, defaulting it to 1. This is possibly because the gradient distribution does not change drastically in strongly convex settings (Johnson & Zhang, 2013). Yet, SVRG's subsequent studies, even those addressing non-convex cases, have neglected the coefficient and formulated their theories based on Equation 1.

Motivated by this, we introduce a time-dependent coefficient vector $\boldsymbol{\alpha}^t \in \mathbb{R}^d$ in SVRG:

$$\boldsymbol{g}_i^t = \nabla f_i(\boldsymbol{\theta}^t) - \boldsymbol{\alpha}^t \odot (\nabla f_i(\boldsymbol{\theta}^{\mathrm{past}}) - \nabla f(\boldsymbol{\theta}^{\mathrm{past}})), \tag{4}$$

where $\odot$ represents the element-wise multiplication.

**Optimal coefficient.** We adopt the same gradient variance definition as Defazio & Bottou (2019) (metric 2 in Table 1) and aim to determine the optimal $\boldsymbol{\alpha}^{t*}$ that minimizes it at each iteration. Specifically, our objective is to minimize the sum of variances across each component of $\boldsymbol{g}_i^t$. Let $k$ index the $k$-th component $\alpha_k^{t*}$ and the $k$-th component of the gradient $\nabla f_{\cdot,k}(\cdot)$. For clarity, we omit the mini-batch index $i$. This can be formally expressed as follows:

$$\min_{\boldsymbol{\alpha}^t} \sum_{k=1}^d \mathrm{Var}(g_{\cdot,k}^t) = \sum_{k=1}^d \min_{\alpha_k^t} \mathrm{Var}(g_{\cdot,k}^t). \tag{5}$$

We can switch the order of minimization and summation in Equation 5 because the variance of the $k$-th component of the gradient only depends on the $k$-th component of the coefficient. Applying Equation 3 yields the optimal coefficient $\alpha_k^{t*}$:

$$\alpha_k^{t*} = \frac{\mathrm{Cov}(\nabla f_{\cdot,k}(\boldsymbol{\theta}^{\mathrm{past}}), \nabla f_{\cdot,k}(\boldsymbol{\theta}^t))}{\mathrm{Var}(\nabla f_{\cdot,k}(\boldsymbol{\theta}^{\mathrm{past}}))} = \rho(\nabla f_{\cdot,k}(\boldsymbol{\theta}^{\mathrm{past}}), \nabla f_{\cdot,k}(\boldsymbol{\theta}^t))\frac{\sigma(\nabla f_{\cdot,k}(\boldsymbol{\theta}^t))}{\sigma(\nabla f_{\cdot,k}(\boldsymbol{\theta}^{\mathrm{past}}))}. \tag{6}$$

A stronger correlation between the snapshot and model gradients leads to a larger optimal coefficient.

For small networks like MLP-4, calculating the optimal coefficient at each iteration is feasible by gathering all mini-batch gradients for both the current and snapshot models. For larger networks, however, this method becomes impractical; we will address this challenge later in the paper.

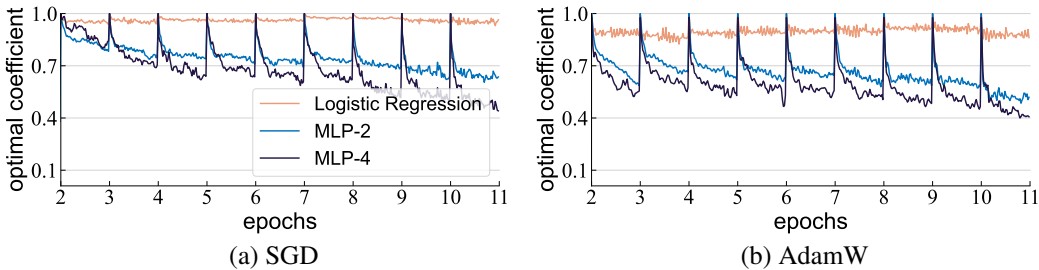

| (a) SGD | (b) AdamW |

Figure 4: **Optimal coefficient.** At the start of each epoch, a snapshot is taken. Consequently, the optimal coefficient initiates at a value of 1 and results in a periodic upward jump.

**Observations on optimal coefficient.** To explore how the optimal coefficient evolves in a normal training setting, we train 1, 2, and 4-layer MLPs (Logistic Regression, MLP-2, and MLP-4) using SGD and AdamW (Loshchilov & Hutter, 2019) on CIFAR-10 *without using SVRG*. Given the small size of these models, we can analytically compute the optimal coefficient at each iteration. We plot its mean value over all indices $k$ in Figure 4. We can make two notable observations as below.

*Observation 1: a deeper model has a smaller optimal coefficient.* For Logistic Regression, the optimal coefficient remains relatively stable, hovering near 1. For MLP-2, the coefficient deviates from 1, dropping to about 0.6. For MLP-4, it decreases more sharply, reaching approximately 0.4.

*Observation 2: the average optimal coefficient of a deeper model in each epoch generally decreases as training progresses.* This suggests that each epoch's average correlation between snapshot gradients and model gradients ($\rho(\nabla f_{\cdot,k}(\boldsymbol{\theta}^{\mathrm{past}}), \nabla f_{\cdot,k}(\boldsymbol{\theta}^t))$ in Equation 6) decreases as the model becomes better trained. We further analyze this decreasing trend of the correlation term in Appendix D.3.

These observations shed light on why the standard SVRG struggles to reduce gradient variance or training loss in later training stages (Figure 3). A default coefficient of 1 proves to be too high, and the weakening correlation between snapshot and model gradients necessitates a smaller coefficient. Without a suitable coefficient, gradient variance may increase, leading to oscillations in SGD.

**Optimal coefficient's effect on gradient variance.** We evaluate whether optimal coefficient can make SVRG more effective in reducing gradient variance. Specifically, we use SVRG with optimal coefficient to train an MLP-4 by computing optimal coefficient (Equation 6) and adjusting the gradient (Equation 4) at each iteration. In Figure 5, we compare SVRG with optimal coefficient to an SGD baseline. Using the optimal coefficient enables SVRG to reduce gradient variance in the early stages of training without uplifting it later. This yields a consistently lower training loss than the baseline.

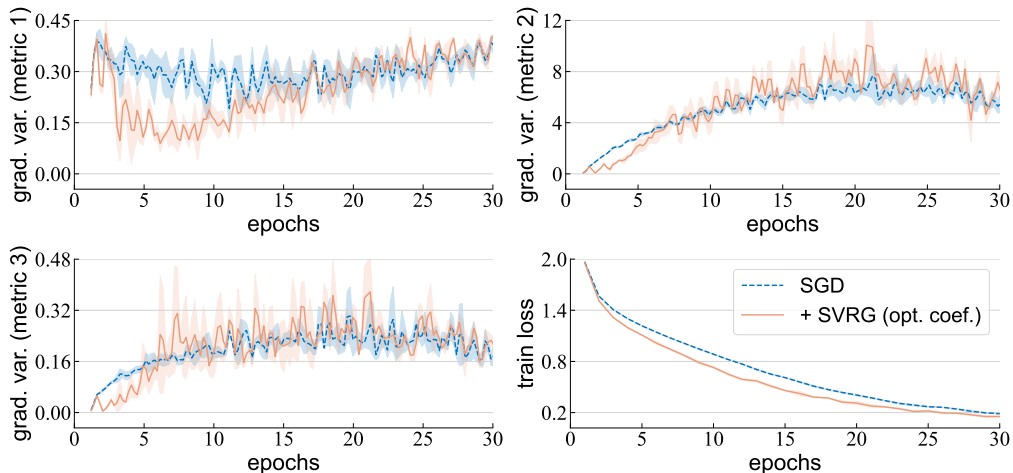

Figure 5: **SVRG with optimal coefficient on MLP-4.** SVRG with the optimal coefficient reduces gradient variance stably and achieves a lower training loss than the baseline SGD.

## 4 $\alpha$-SVRG

From our analysis above, it becomes clear that the best coefficient for SVRG is not necessarily 1 for deep neural networks. However, computing the optimal coefficient at each iteration would result in a complexity of full-batch gradient descent. This approach quickly becomes impractical for larger networks like ResNet (He et al., 2016). In this section, we show how using a preset schedule of $\alpha$ values can achieve a similar effect of using the computed optimal coefficients.

**$\alpha$-SVRG.** Given the decreasing trend (Figure 4) and the computational challenge, we propose to apply a linearly decreasing scalar coefficient for SVRG, starting from an initial value $\alpha_0$ and decreasing to 0. This is our main method in this paper. We name it $\alpha$-SVRG. More results of enabling $\alpha$-SVRG only during the early stage of training are in Appendix C.4. The pseudocode for $\alpha$-SVRG with SGD and AdamW as base optimizers is provided in Appendix G.

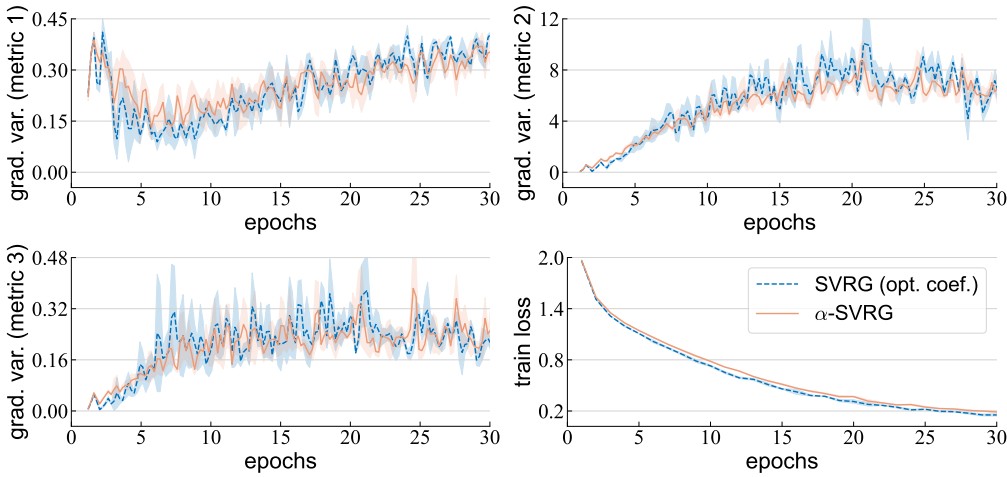

Figure 6: **$\alpha$-SVRG on MLP-4.** $\alpha$-SVRG behaves similarly to SVRG with optimal coefficient.

To evaluate how well $\alpha$-SVRG matches SVRG with optimal coefficient, we train an MLP-4 using $\alpha$-SVRG and compare it to SVRG with optimal coefficient. For all experiments in this section, we set $\alpha_0 = 0.5$. The results are presented in Figure 6. Interestingly, $\alpha$-SVRG exhibits a gradient variance trend that is not much different from SVRG with optimal coefficient. Similarly, the training loss of $\alpha$-SVRG is only marginally larger than that of SVRG with optimal coefficient.

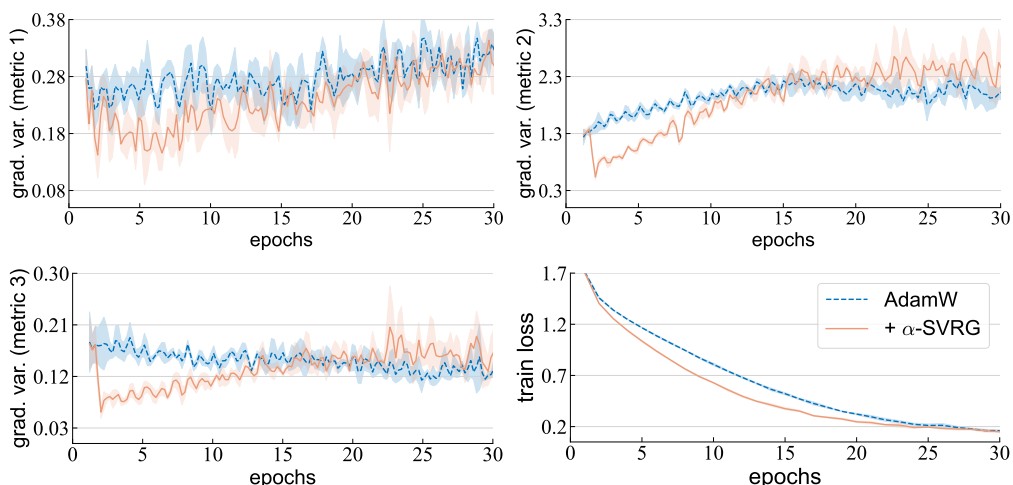

Figure 7: **$\alpha$-SVRG with AdamW on MLP-4.** $\alpha$-SVRG can lower the gradient variance at the first 10 epochs, leading to a faster convergence than the baseline AdamW.

**$\alpha$-SVRG with AdamW.** Since AdamW (Loshchilov & Hutter, 2019) is a widely used optimizer in modern neural network training, we assess the performance of $\alpha$-SVRG with AdamW. We change the base optimizer in $\alpha$-SVRG to AdamW and use it to train an MLP-4 on CIFAR-10. We compare $\alpha$-SVRG to a baseline using only AdamW. As shown in Figure 7, $\alpha$-SVRG has a noticeable gradient variance reduction initially and achieves a consistent lower training loss for MLP-4 than the baseline.

**$\alpha$-SVRG on deeper networks.** We further study the effectiveness of $\alpha$-SVRG with AdamW on real-world neural architectures, moving beyond simple MLPs. To this end, we train a modern ConvNet architecture, ConvNeXt-Femto (Liu et al., 2022; Wightman, 2019), on CIFAR-10 using the default AdamW optimizer. We compare $\alpha$-SVRG to the baseline using vanilla AdamW in Figure 8. $\alpha$-SVRG can reduce gradient variance during the first 10 epochs (zoom-in plot of Figure 8) and then maintain it at the same level as the baseline. As a result, the training loss of $\alpha$-SVRG converges much faster than the baseline. This demonstrates the potential of $\alpha$-SVRG in optimizing more complex models. We further explore this with additional experiments next.

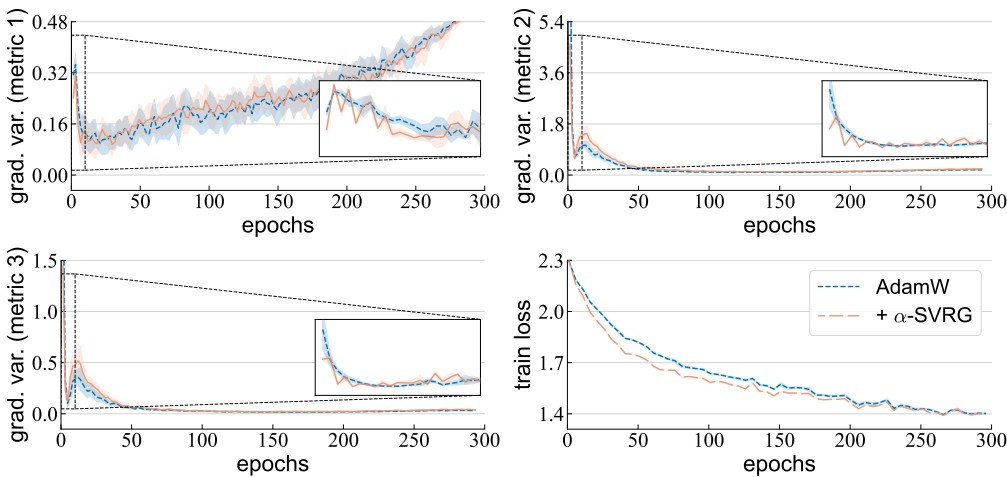

Figure 8: **$\alpha$-SVRG on ConvNeXt-Femto**. $\alpha$-SVRG can reduce the gradient variance for ConvNeXt-Femto during the first 10 epochs (zoom-in plot) without increasing it later on. Consequently, $\alpha$-SVRG can decrease the training loss at a faster rate than the baseline AdamW.

# 5  EXPERIMENTS

## 5.1  SETTINGS

**Datasets.** We evaluate $\alpha$-SVRG using ImageNet-1K classification (Deng et al., 2009) as well as smaller image classification datasets: CIFAR-100 (Krizhevsky, 2009), Pets (Parkhi et al., 2012), Flowers (Nilsback & Zisserman, 2008), STL-10 (Coates et al., 2011), Food-101 (Bossard et al., 2014), DTD (Cimpoi et al., 2014), SVHN (Netzer et al., 2011), and EuroSAT (Helber et al., 2019).

**Models.** We use recently proposed vision models on ImageNet-1K, categorized into two groups: (1) smaller models with 5-19M parameters, including ConvNeXt-F (Wightman, 2019; Liu et al., 2022), ViT-T/16 (Dosovitskiy et al., 2021), Swin-F (Liu et al., 2021b), and Mixer-S/32 (Tolstikhin et al., 2021); (2) larger models featuring 86M and 89M parameters: ViT-B/16 and ConvNeXt-B. ConvNeXt-F is also evaluated on all smaller image classification datasets.

**Training.** We report both final epoch training loss and top-1 validation accuracy. Our basic training setting follows ConvNeXt (Liu et al., 2022), which uses AdamW. Both SVRG and $\alpha$-SVRG also use AdamW as the base optimizer. On small datasets, we choose the best $\alpha_0$ from $\{0.5, 0.75, 1\}$. We find the coefficient is robust and does not require extensive tuning. Therefore, for ImageNet-1K, we set $\alpha_0$ to 0.75 for smaller models and 0.5 for larger ones. Other training settings for $\alpha$-SVRG remain the same as the baseline. Further experimental settings can be found in Appendix B.

| | ConvNeXt-F | | ViT-T | | Swin-F | | Mixer-S | | ViT-B | | ConvNeXt-B | |
|---|---|---|---|---|---|---|---|---|---|---|---|---|
| | training loss | | | | | | | | | | | |
| AdamW | 3.487 | - | 3.443 | - | 3.427 | - | 3.149 | - | 2.817 | - | 2.644 | - |
| + SVRG | 3.505 | ↑.018 | 3.431 | ↓.012 | **3.389** | ↓.038 | 3.172 | ↑.023 | 3.309 | ↑.492 | 3.113 | ↑.469 |
| + $\alpha$-SVRG | **3.467** | ↓.020 | **3.415** | ↓.028 | 3.392 | ↓.035 | **3.097** | ↓.052 | **2.806** | ↓.011 | **2.642** | ↓.002 |
| | validation accuracy | | | | | | | | | | | |
| AdamW | 76.0 | - | 73.9 | - | 74.3 | - | **76.4** | - | **81.6** | - | **83.7** | - |
| + SVRG | 75.7 | ↓0.3 | **74.3** | ↑0.4 | 74.3 | ↑0.0 | 74.5 | ↓1.9 | 78.0 | ↓3.6 | 80.8 | ↓2.9 |
| + $\alpha$-SVRG | **76.3** | ↑0.3 | 74.2 | ↑0.3 | **74.8** | ↑0.5 | 76.1 | ↓0.3 | **81.6** | ↑0.0 | 83.1 | ↓0.6 |

Table 2: **Results on ImageNet-1K.** The standard SVRG increases the training loss for most models, whereas $\alpha$-SVRG consistently decreases it for all models.

| | CIFAR-100 | | Pets | | Flowers | | STL-10 | | Food-101 | | DTD | | SVHN | | EuroSAT | |
|---|---|---|---|---|---|---|---|---|---|---|---|---|---|---|---|---|
| | training loss | | | | | | | | | | | | | | | |
| AdamW | 2.66 | - | 2.20 | - | 2.40 | - | 1.64 | - | 2.45 | - | 1.98 | - | 1.59 | - | 1.25 | - |
| + SVRG | 2.94 | ↑0.28 | 3.42 | ↑1.22 | 2.26 | ↓0.14 | 1.90 | ↑0.26 | 3.03 | ↑0.58 | 2.01 | ↑0.03 | 1.64 | ↑0.05 | 1.25 | 0.00 |
| + $\alpha$-SVRG | **2.62** | ↓0.04 | **1.96** | ↓0.24 | **2.16** | ↓0.24 | **1.57** | ↓0.07 | **2.42** | ↓0.03 | **1.83** | ↓0.15 | **1.57** | ↓0.02 | **1.23** | ↓0.02 |
| | validation accuracy | | | | | | | | | | | | | | | |
| AdamW | 81.0 | - | 72.8 | - | 80.8 | - | 82.3 | - | **85.9** | - | 57.9 | - | 94.9 | - | 98.1 | - |
| + SVRG | 78.2 | ↓2.8 | 17.6 | ↓55.2 | 82.6 | ↑1.8 | 65.1 | ↓17.2 | 79.6 | ↓6.3 | 57.8 | ↓0.1 | 95.7 | ↑0.8 | 97.9 | ↓0.2 |
| + $\alpha$-SVRG | **81.4** | ↑0.4 | **77.8** | ↑5.0 | **83.3** | ↑2.5 | **84.0** | ↑1.7 | **85.9** | ↑0.0 | **61.8** | ↑3.9 | **95.8** | ↑0.9 | **98.2** | ↑0.1 |

Table 3: **Results on smaller classification datasets.** While the standard SVRG mostly hurts the performance, $\alpha$-SVRG decreases the training loss and increases the validation accuracy.

## 5.2 RESULTS

Table 2 presents the results of training various models on ImageNet-1K. The standard SVRG often increases the training loss, especially for larger models. In contrast, $\alpha$-SVRG decreases the training loss for both smaller and larger models. This also supports our earlier finding that deeper models benefit from lower coefficient values, and using a default coefficient of 1 impedes convergence.

Table 3 displays the results of training ConvNeXt-F on various smaller datasets. The standard SVRG generally elevates the training loss and impairs the generalization. On the contrary, $\alpha$-SVRG lowers the training loss and improves the validation accuracy across all small datasets. We have provided additional experiment results to demonstrate $\alpha$-SVRG's effectiveness in Appendix C.

Note that a lower training loss in $\alpha$-SVRG does not always lead to better generalization. For smaller models, a lower training loss usually directly translates to a higher validation accuracy. In larger models (Mixer-S, ViT-B, and ConvNeXt-B), additional adjustments to regularization strength may be needed for better generalization. This is out of scope for $\alpha$-SVRG as an optimization method but warrants future research on co-adapting optimization and regularization.

## 5.3 ANALYSIS

We analyze various components in $\alpha$-SVRG. In the following experiments, we use an initial value $\alpha_0 = 0.5$ and ConvNeXt-F on STL-10 as the default setting. Because the standard SVRG is ineffective as discussed above, we omit it and only compare $\alpha$-SVRG to an AdamW baseline.

**Coefficient value.** We investigate the impact of the initial value of the coefficient $\alpha_0$ for $\alpha$-SVRG. We vary it between 0 and 1 and observe its effect on the training loss. The results are presented in Figure 9. The favorable range for initial values in $\alpha$-SVRG is quite broad, ranging from 0.2 to 0.9. This robustness indicates $\alpha$-SVRG requires minimal tuning in the practical setting.

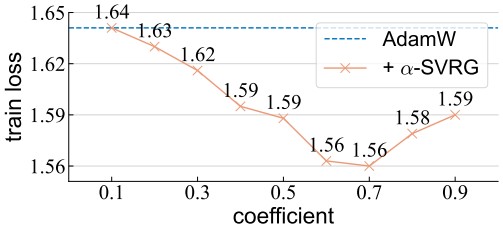
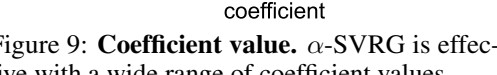
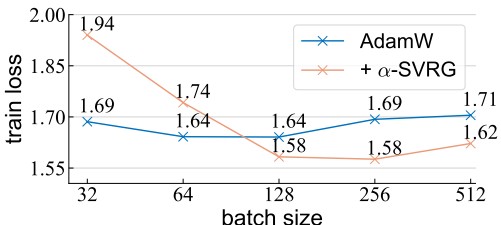

Figure 9: **Coefficient value.** $\alpha$-SVRG is effective with a wide range of coefficient values.

Figure 10: **Batch size.** $\alpha$-SVRG's effectiveness is observed for larger batch sizes.

**Batch size.** Since the batch size controls the variance among mini-batch data, we change the batch size to understand how it affects $\alpha$-SVRG. We also scale the learning rate linearly (Goyal et al., 2017). The default batch size is 128. In Figure 10, we can see that $\alpha$-SVRG leads to a lower training loss when the batch size is larger, but it is worse than the baseline when the batch size is smaller. This may stem from the weakening correlation between snapshot gradients and model gradients as the batch size decreases. Therefore, a sufficiently large batch size is essential for $\alpha$-SVRG.

**Coefficient schedule.** By default, our $\alpha$-SVRG uses a linearly decreasing schedule to adjust the coefficient. Below we explore other schedules and illustrate them in Figure 11. Global schedules only decay the coefficient *across epochs and keep as a constant within an epoch*. In contrast, double schedules also model the local decay in each epoch (Figure 4) by initiating the coefficient at 1 and decreasing to an ending value specified by the global decay. More details on them are in Appendix E.

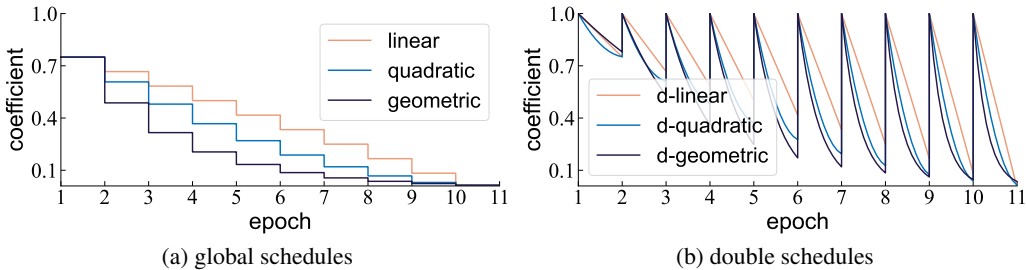

(a) global schedules          (b) double schedules

Figure 11: **Different coefficient schedules with $\alpha_0 = 0.75$.** Each global schedule (left) maintains a static coefficient within an epoch and applies a coefficient decay only at the end of each epoch. In contrast, each double schedule (right) also adjusts the coefficient within an epoch.

Table 4 presents the results of $\alpha$-SVRG using each schedule. $\alpha$-SVRG with double schedules surprisingly have a higher training loss than the AdamW baseline (1.64). This is possibly because the coefficient within an epoch sometimes overestimates the optimal coefficient and therefore increases gradient variance. In contrast, $\alpha$-SVRG with global schedules consistently achieves a lower training loss than the baseline (1.64) regardless of the choice of any initial coefficient.

| train loss | linear | quadratic | geometric | d-linear | d-quadratic | d-geometric |
|---|---|---|---|---|---|---|
| $\alpha_0 = 0.5$ | **1.59** | 1.61 | 1.62 | 2.07 | 1.97 | 1.81 |
| $\alpha_0 = 0.75$ | **1.57** | 1.58 | 1.58 | 2.07 | 2.00 | 1.93 |
| $\alpha_0 = 1$ | 1.57 | **1.56** | 1.57 | 2.00 | 1.97 | 1.88 |

Table 4: **Schedules. Bold** indicates the lowest training loss among different schedules using the same initial coefficient (row). $\alpha$-SVRG with global schedules outperforms that with double schedules.

**Inner loop size.** The inner loop size specifies the number of iterations between two consecutive snapshot captures. We vary it from 1 to 312 iterations to understand its effect on $\alpha$-SVRG. The default value is 39 iterations (one epoch). Figure 12 illustrates $\alpha$-SVRG has a lower training loss than the baseline even with a larger inner loop size, where the snapshot is relatively distant from the current model. On the other hand, a smaller inner loop size results in a lower training loss but requires additional training time, as a full gradient must be calculated each time a snapshot is taken.

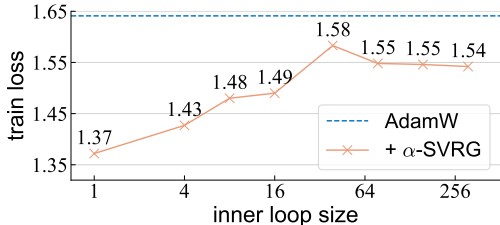

Figure 12: **Inner loop size.** Although a greater inner loop size leads to a weakening correlation between the model gradients and the snapshot gradients, $\alpha$-SVRG can still lower training loss.

## 6  RELATED WORK

**Variance reduction in optimization.** There are a range of methods aiming at reducing gradient variance by directly modifying stochastic gradient. Initial works (Johnson & Zhang, 2013; Schmidt et al., 2016) focus on simple convex settings. Subsequent research enhances these methods (Defazio et al., 2014a; Mairal, 2015; Lin et al., 2015; Defazio, 2016; Allen-Zhu, 2017; Lin et al., 2018) or handles finite sums in non-convex landscapes (Allen-Zhu & Hazan, 2016; Nguyen et al., 2017; Fang et al., 2018; Li & Li, 2018; Cutkosky & Orabona, 2019; Elibol et al., 2020; Zhou et al., 2020; Kavis et al., 2022). For these methods, we either need to store all gradient with respect to each individual data point (Defazio et al., 2014b; Shalev-Shwartz & Zhang, 2013; Li et al., 2021) or calculate full gradient periodically (Johnson & Zhang, 2013; Fang et al., 2018). Gower et al. (2020) provide a comprehensive review for variance reduction methods. While these studies focus on theories of SVRG, we primarily explore the practical utility of SVRG for real-world neural networks.

One of the most relevant works to us is MARS (Yuan et al., 2024), which also demonstrates that using a coefficient helps the variance reduction methods optimize modern neural networks. In contrast, our work studies how a coefficient makes SVRG effective through step-by-step controlled experiments.

**Implicit variance reduction.** Apart from methods that explicitly adjust the gradient, there are variance reduction techniques that implicitly reduce gradient variance through other means. A variety of optimizers (Zeiler, 2012; Kingma & Ba, 2015; Dozat, 2016; Lydia & Francis, 2019; Loshchilov & Hutter, 2019; Liu et al., 2021a; 2024; Chen et al., 2023) utilize momentum to mitigate gradient variance. They achieve this by averaging past gradients exponentially, thus stabilizing subsequent updates. Lookahead optimizer (Zhang et al., 2019) reduces gradient variance by only updating the model once every $k$ iterations. Dropout (Hinton et al., 2012) is also found to reduce gradient variance and better optimize models when used at early training (Liu et al., 2023).

## 7  CONCLUSION

Over the past decade, SVRG has been a method with a significant impact on the theory of optimization. In this work, we explore the effectiveness of SVRG in training real-world neural networks. Our key insight is the optimal strength for the variance reduction term in SVRG is not necessarily 1. It should be lower for deeper networks and decrease as training advances. This motivates us to introduce $\alpha$-SVRG: applying a linearly decreasing coefficient $\alpha$ to SVRG. $\alpha$-SVRG leads to a steady reduction in gradient variance and optimizes models better. Our experiments show that $\alpha$-SVRG consistently achieves a lower training loss compared to both baseline and the standard SVRG. Our results motivate further research of variance reduction methods in neural networks training.

**Acknowledgement.** We would like to thank Kaiming He, Aaron Defazio, Zeyuan Allen-Zhu, Kirill Vishniakov, Huijin Ou, Jiayi Xu, Shuyi Wang, and Zekai Wang for valuable discussions and feedback.

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

# APPENDIX

## A DERIVATION OF THE OPTIMAL COEFFICIENT

We present the full derivation of the optimal coefficient for control variates:

$$\min_{\alpha} \text{Var}(X^*) = \min_{\alpha} \text{Var}(X - \alpha Y) \tag{7}$$

$$= \min_{\alpha} \text{Var}(X) - 2\alpha \text{Cov}(X, Y) + \alpha^2 \text{Var}(Y). \tag{8}$$

Differentiating the objective with respect to $\alpha$, we can determine the optimal coefficient $\alpha^*$:

$$2\alpha \text{Var}(Y) - 2\text{Cov}(X, Y) = 0, \tag{9}$$

$$\implies \alpha^* = \frac{\text{Cov}(X, Y)}{\text{Var}(Y)}. \tag{10}$$

Lastly, we can plug the definition of the correlation coefficient:

$$\rho(X, Y) = \frac{\text{Cov}(X, Y)}{\sigma(X)\sigma(Y)} \tag{11}$$

into the optimal coefficient and rewrite Equation 10 as:

$$\alpha^* = \rho(X, Y)\frac{\sigma(X)}{\sigma(Y)}. \tag{12}$$

## B EXPERIMENTAL SETTINGS

**Training recipe.** Table 5 outlines our training recipe. It is based on the setting in ConvNeXt (Liu et al., 2022). For all experiments, the base learning rate is set at 4e-3, except for training ConvNeXt-F on ImageNet-1K using $\alpha$-SVRG, where increasing it to 8e-3 reduces training loss very much.

| config | value |
|---|---|
| weight init | trunc. normal (0.2) |
| optimizer | AdamW |
| base learning rate | 4e-3 |
| weight decay | 0.05 |
| optimizer momentum | $\beta_1, \beta_2 = 0.9, 0.999$ |
| learning rate schedule | cosine decay |
| warmup schedule | linear |
| randaugment (Cubuk et al., 2020) | $(9, 0.5)$ |
| mixup (Zhang et al., 2018) | 0.8 |
| cutmix (Yun et al., 2019) | 1.0 |
| random erasing (Zhong et al., 2020) | 0.25 |
| label smoothing (Szegedy et al., 2016) | 0.1 |

Table 5: **Our basic training recipe,** adapted from ConvNeXt (Liu et al., 2022).

**Hyperparameters.** Table 6 lists the batch size, warmup epochs, and training epochs for each dataset. Note that these hyperparameters selections are done on the AdamW baseline. For each dataset, we

| | CIFAR-100 | Pets | Flowers | STL-10 | Food101 | DTD | SVHN | EuroSAT | ImageNet-1K |
|---|---|---|---|---|---|---|---|---|---|
| batch size | 1024 | 128 | 128 | 128 | 1024 | 128 | 1024 | 512 | 4096 |
| warmup epochs | 50 | 100 | 100 | 50 | 50 | 100 | 20 | 40 | 50 |
| training epochs | 300 | 600 | 600 | 300 | 300 | 600 | 100 | 200 | 300 |

Table 6: **Hyperparameter setting.**

set the batch size proportional to its total size and tune the training epochs to achieve reasonable performance. The warmup epochs are set to one-fifth or one-sixth of the total training epochs.

We do not use stochastic depth (Huang et al., 2016) on small models. For larger models, we adhere to the original work (Dosovitskiy et al., 2021; Liu et al., 2022), using a stochastic depth rate of 0.4 for ViT-B and 0.5 for ConvNeXt-B. In these models, we maintain a consistent stochastic pattern between the current model and the snapshot at each iteration (Defazio & Bottou, 2019).

## C ADDITIONAL RESULTS OF $\alpha$-SVRG

In this section, we provide additional experiment results to demonstrate the effectiveness of $\alpha$-SVRG. This includes full results of $\alpha$-SVRG with different initial coefficients $\alpha_0$ on small image classification datasets (Appendix C.1), training with three random seeds (Appendix C.2), a full learning rate search (Appendix C.3), and applying $\alpha$-SVRG only in the early training stage (Appendix C.4).

### C.1 DIFFERENT INITIAL COEFFICIENTS

Table 7 presents the performance of ConvNeXt-F trained with $\alpha$-SVRG using different initial coefficients $\alpha_0$ on various image classification datasets. $\alpha$-SVRG consistently reduces the training loss of ConvNeXt-F and enhances the validation accuracy on most datasets, regardless of the choice of initial coefficient $\alpha_0$. This demonstrates the robustness of $\alpha$-SVRG to the initial coefficient.

| | CIFAR-100 | | Pets | | Flowers | | STL-10 | |
|---|---|---|---|---|---|---|---|---|
| AdamW | 2.659 | - | 2.203 | - | 2.400 | - | 1.641 | - |
| + SVRG | 2.937 | ↑0.278 | 3.424 | ↑1.221 | 2.256 | ↓0.144 | 1.899 | ↑0.258 |
| + $\alpha$-SVRG ($\alpha_0 = 0.5$) | **2.622** | ↓0.037 | 1.960 | ↓0.243 | 2.265 | ↓0.135 | 1.583 | ↓0.058 |
| + $\alpha$-SVRG ($\alpha_0 = 0.75$) | 2.646 | ↓0.013 | 2.004 | ↓0.199 | **2.162** | ↓0.238 | **1.568** | ↓0.073 |
| + $\alpha$-SVRG ($\alpha_0 = 1$) | 2.712 | ↑0.053 | **1.994** | ↓0.209 | 2.259 | ↓0.141 | 1.573 | ↓0.068 |
| | Food-101 | | DTD | | SVHN | | EuroSAT | |
| AdamW | 2.451 | - | 1.980 | - | 1.588 | - | 1.247 | - |
| + SVRG | 3.026 | ↑0.575 | 2.009 | ↑0.029 | 1.639 | ↑0.051 | 1.249 | ↑0.002 |
| + $\alpha$-SVRG ($\alpha_0 = 0.5$) | **2.423** | ↓0.028 | 1.865 | ↓0.115 | **1.572** | ↓0.016 | 1.243 | ↓0.004 |
| + $\alpha$-SVRG ($\alpha_0 = 0.75$) | 2.461 | ↑0.010 | 1.829 | ↓0.151 | 1.573 | ↓0.015 | 1.237 | ↓0.010 |
| + $\alpha$-SVRG ($\alpha_0 = 1$) | 2.649 | ↑0.198 | **1.790** | ↓0.190 | 1.585 | ↓0.003 | **1.230** | ↓0.017 |

(a) **training loss**

| | CIFAR-100 | | Pets | | Flowers | | STL-10 | |
|---|---|---|---|---|---|---|---|---|
| AdamW | 81.0 | - | 72.8 | - | 80.8 | - | 82.3 | - |
| + SVRG | 78.2 | ↓2.8 | 17.6 | ↓55.2 | 82.6 | ↑1.8 | 65.1 | ↓17.2 |
| + $\alpha$-SVRG ($\alpha_0 = 0.5$) | **81.4** | ↑0.4 | **77.8** | ↑5.0 | **83.3** | ↑2.5 | **83.5** | ↑1.2 |
| + $\alpha$-SVRG ($\alpha_0 = 0.75$) | 80.6 | ↓0.4 | 76.7 | ↑3.9 | 82.6 | ↑1.8 | 84.0 | ↑1.7 |
| + $\alpha$-SVRG ($\alpha_0 = 1$) | 80.0 | ↓1.0 | 77.3 | ↑4.5 | 81.9 | ↑1.1 | 84.0 | ↑1.7 |
| | Food-101 | | DTD | | SVHN | | Euro | |
| AdamW | **85.9** | - | 57.9 | - | 94.9 | - | 98.1 | - |
| + SVRG | 79.6 | ↓6.3 | 57.8 | ↓0.1 | 95.7 | ↑0.8 | 97.9 | ↓0.2 |
| + $\alpha$-SVRG ($\alpha_0 = 0.5$) | **85.9** | ↑0.0 | 57.0 | ↓0.9 | 95.4 | ↑0.5 | **98.2** | ↑0.1 |
| + $\alpha$-SVRG ($\alpha_0 = 0.75$) | 85.0 | ↓0.9 | 60.3 | ↑2.4 | 95.7 | ↑0.8 | **98.2** | ↑0.1 |
| + $\alpha$-SVRG ($\alpha_0 = 1$) | 83.8 | ↓2.1 | **61.8** | ↑3.9 | **95.8** | ↑0.9 | **98.2** | ↑0.1 |

(b) **validation accuracy**

Table 7: **Results on smaller classification datasets with different initial coefficients.** While SVRG negatively affects performance on most of these datasets, $\alpha$-SVRG consistently reduces the training loss and improves the validation accuracy for almost any initial coefficient on each dataset.

## C.2 STANDARD DEVIATION RESULTS

Here we run the AdamW baseline and $\alpha$-SVRG in Table 3 with 3 random seeds. Table 8 presents the results. $\alpha$-SVRG decreases the mean training loss and improves the mean validation accuracy. The mean difference is usually larger than one standard deviation, indicating the reliability of $\alpha$-SVRG.

| | CIFAR-100 | Pets | Flowers | STL-10 |
|---|---|---|---|---|
| AdamW | $2.645 \pm 0.013$ | $2.326 \pm 0.088$ | $2.436 \pm 0.038$ | $1.660 \pm 0.017$ |
| + $\alpha$-SVRG | $\mathbf{2.606} \pm 0.017$ | $\mathbf{2.060} \pm 0.071$ | $\mathbf{2.221} \pm 0.042$ | $\mathbf{1.577} \pm 0.022$ |
| | Food-101 | DTD | SVHN | EuroSAT |
| AdamW | $2.478 \pm 0.021$ | $2.072 \pm 0.066$ | $1.583 \pm 0.005$ | $1.259 \pm 0.017$ |
| + $\alpha$-SVRG | $\mathbf{2.426} \pm 0.007$ | $\mathbf{1.896} \pm 0.075$ | $\mathbf{1.572} \pm 0.011$ | $\mathbf{1.239} \pm 0.016$ |

(a) **training loss**

| | CIFAR-100 | Pets | Flowers | STL-10 |
|---|---|---|---|---|
| AdamW | $81.02 \pm 0.07$ | $70.61 \pm 1.55$ | $80.33 \pm 1.01$ | $80.80 \pm 1.46$ |
| + $\alpha$-SVRG | $\mathbf{81.07} \pm 0.22$ | $\mathbf{76.37} \pm 1.06$ | $\mathbf{84.15} \pm 1.15$ | $\mathbf{83.65} \pm 0.92$ |
| | Food-101 | DTD | SVHN | EuroSAT |
| AdamW | $85.29 \pm 0.47$ | $56.21 \pm 1.19$ | $94.29 \pm 0.67$ | $97.91 \pm 0.12$ |
| + $\alpha$-SVRG | $\mathbf{85.45} \pm 0.43$ | $\mathbf{61.44} \pm 0.35$ | $\mathbf{94.94} \pm 0.60$ | $\mathbf{98.13} \pm 0.07$ |

(b) **validation accuracy**

Table 8: **Results of $\alpha$-SVRG with AdamW with standard deviation.**

In Section 5, we primarily use AdamW as the base optimizer to study $\alpha$-SVRG. Here we switch the base optimizer in $\alpha$-SVRG from AdamW to SGD. Specifically, we train a ResNet-18 (He et al., 2016), which by default uses SGD to train, on the small image classification datasets. Following He et al. (2016), we use an initial learning rate of 0.1, which is divided by 10 when the error plateaus, a momentum of 0.9, and a weight decay of 1e-4. Other settings in training recipe remain the same as Table 5. Table 10 details other hyperparameters, such as batch size and training epochs.

| | CIFAR-100 | Pets | Flowers | STL-10 |
|---|---|---|---|---|
| SGD | $3.118 \pm 0.035$ | $2.706 \pm 0.095$ | $2.822 \pm 0.058$ | $1.763 \pm 0.032$ |
| + $\alpha$-SVRG | $\mathbf{3.087} \pm 0.011$ | $\mathbf{2.655} \pm 0.134$ | $\mathbf{2.699} \pm 0.049$ | $\mathbf{1.725} \pm 0.043$ |
| | Food-101 | DTD | SVHN | EuroSAT |
| SGD | $3.424 \pm 0.015$ | $2.589 \pm 0.032$ | $1.789 \pm 0.029$ | $1.449 \pm 0.029$ |
| + $\alpha$-SVRG | $\mathbf{3.397} \pm 0.006$ | $\mathbf{2.543} \pm 0.039$ | $\mathbf{1.764} \pm 0.014$ | $\mathbf{1.412} \pm 0.011$ |

(a) **training loss**

| | CIFAR-100 | Pets | Flowers | STL-10 |
|---|---|---|---|---|
| SGD | $75.43 \pm 0.88$ | $71.25 \pm 4.74$ | $65.92 \pm 4.24$ | $76.09 \pm 1.09$ |
| + $\alpha$-SVRG | $\mathbf{75.93} \pm 0.83$ | $\mathbf{71.89} \pm 4.84$ | $\mathbf{74.98} \pm 3.14$ | $\mathbf{78.55} \pm 2.54$ |
| | Food-101 | DTD | SVHN | EuroSAT |
| SGD | $60.58 \pm 2.00$ | $57.53 \pm 1.25$ | $91.10 \pm 1.31$ | $95.31 \pm 0.46$ |
| + $\alpha$-SVRG | $\mathbf{62.89} \pm 0.87$ | $\mathbf{58.44} \pm 1.05$ | $\mathbf{91.60} \pm 0.59$ | $\mathbf{96.17} \pm 0.17$ |

(b) **validation accuracy**

Table 9: **Results of $\alpha$-SVRG with SGD on smaller datasets with standard deviation.**

We compare $\alpha$-SVRG ($\alpha_0 = 0.5$) equipped by a linear decreasing schedule to the baseline using only SGD. The results, shown in Table 9, are based on the average of 3 runs with different random seeds. As we can see, $\alpha$-SVRG consistently outperforms the SGD baseline across all datasets.

|  | CIFAR-100 | Pets | Flowers | STL-10 | Food101 | DTD | SVHN | EuroSAT |
|---|---|---|---|---|---|---|---|---|
| batch size | 1024 | 128 | 128 | 128 | 1024 | 128 | 1024 | 512 |
| training epochs | 150 | 200 | 150 | 200 | 50 | 200 | 50 | 100 |

Table 10: **Hyperparameter setting for $\alpha$-SVRG with SGD on ResNet-18.**

### C.3 DIFFERENT LEARNING RATES

$\alpha$**-SVRG with AdamW.** In Section 5, we use the same base learning rate of 4e-3 for both AdamW and $\alpha$-SVRG. However, each method's optimal learning rate might be different. Here we sweep the base learning rate over the range {1e-2, 5e-3, 1e-3, 5e-4, 1e-4} for both methods. As shown in Figure 13, $\alpha$-SVRG ($\alpha_0 = 0.5$) can decrease training loss better than vanilla AdamW in most cases.

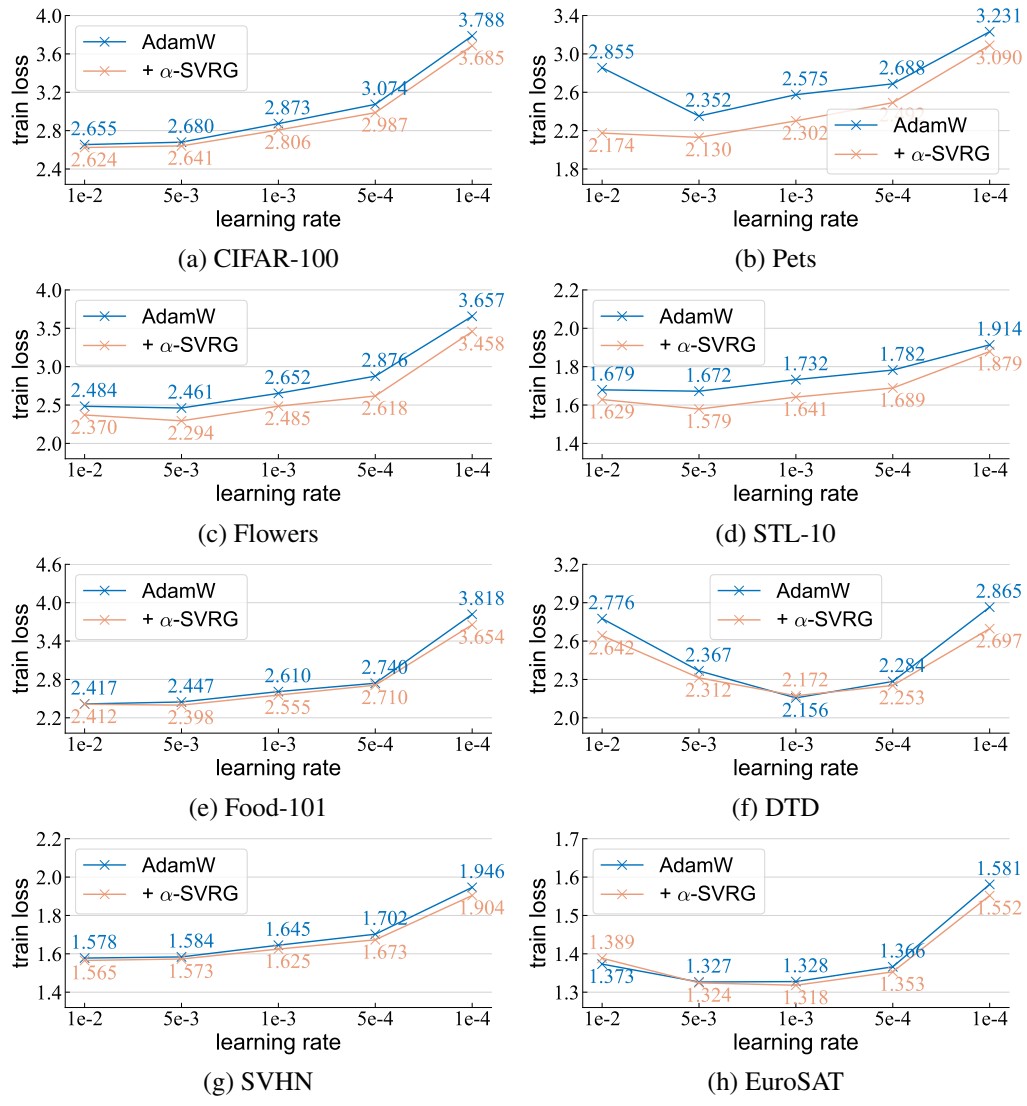

Figure 13: $\alpha$**-SVRG with AdamW at different learning rates**.

$\alpha$**-SVRG with SGD.** We also sweep the base learning rate for the results of Table 9 using SGD as the base optimizer. We compare vanilla SGD to $\alpha$-SVRG ($\alpha_0 = 0.5$). The results are shown in Figure 14. In most learning rates, $\alpha$-SVRG can achieve a lower training loss than SGD.

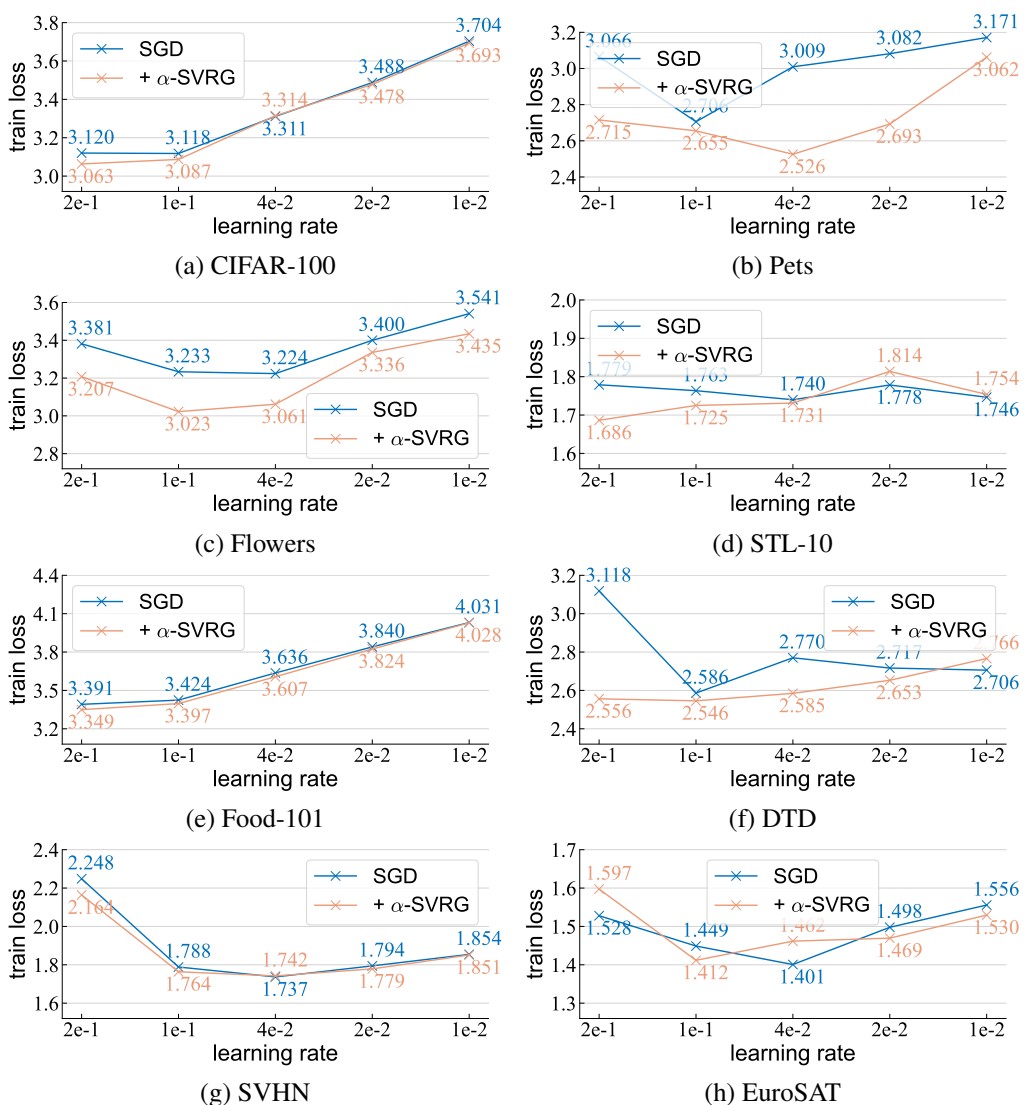

Figure 14: $\alpha$**-SVRG with SGD at different learning rates**.

## C.4 USING $\alpha$-SVRG DURING THE INITIAL PHASES OF TRAINING

Compared with SGD and AdamW, both standard SVRG and $\alpha$-SVRG require computing the snapshot stochastic gradient $\nabla f_i(\theta^{\text{past}})$ and snapshot full gradient $\nabla f(\theta^{\text{past}})$. This leads to about twice the computational cost of the baseline methods. Nevertheless, in Section 4, we find that $\alpha$-SVRG can only reduce gradient variance during the initial training epochs and then maintains at a similar level as the baseline methods. This motivates us to apply $\alpha$-SVRG *only at the early phases of training*, and we refer this approach as early $\alpha$-SVRG.

To evaluate this approach, we use early $\alpha$-SVRG to train ConvNeXt-Femto on various image classification datasets. We use the default linear decay with an initial coefficient $\alpha_0 = 0.75$ to schedule early $\alpha$-SVRG, but $\alpha$-SVRG is only applied during the first 10% of training and is disabled thereafter. Moreover, we add a transition epoch where the coefficient decreases from its original final value to zero. We find this crucial for maintaining the stability of momentum in the base optimizer.

| | CIFAR-100 | | Pets | | Flowers | | STL-10 | |
|---|---|---|---|---|---|---|---|---|
| AdamW | 2.659 | - | 2.203 | - | 2.400 | - | 1.641 | - |
| + $\alpha$-SVRG | 2.646 | ↓0.013 | 2.004 | ↓0.199 | 2.162 | ↓0.238 | 1.568 | ↓0.073 |
| + early $\alpha$-SVRG | 2.644 | ↓0.015 | 2.190 | ↓0.013 | 2.328 | ↓0.072 | 1.616 | ↓0.025 |
| | Food-101 | | DTD | | SVHN | | EuroSAT | |
| AdamW | 2.451 | - | 1.980 | - | 1.588 | - | 1.247 | - |
| + $\alpha$-SVRG | 2.461 | ↑0.010 | 1.829 | ↓0.151 | 1.573 | ↓0.015 | 1.237 | ↓0.010 |
| + early $\alpha$-SVRG | 2.444 | ↓0.007 | 1.918 | ↓0.062 | 1.583 | ↓0.005 | 1.240 | ↓0.007 |

(a) **training loss**

| | CIFAR-100 | | Pets | | Flowers | | STL-10 | |
|---|---|---|---|---|---|---|---|---|
| baseline | 81.0 | - | 72.8 | - | 80.8 | - | 82.3 | - |
| + $\alpha$-SVRG | 80.6 | ↓0.4 | 76.7 | ↑3.9 | 82.6 | ↑1.8 | 84.0 | ↑1.7 |
| + early $\alpha$-SVRG | 81.0 | ↑0.0 | 74.6 | ↑1.8 | 83.6 | ↑2.8 | 82.8 | ↑0.5 |
| | Food-101 | | DTD | | SVHN | | Euro | |
| baseline | 85.9 | - | 57.9 | - | 94.9 | - | 98.1 | - |
| + $\alpha$-SVRG | 85.0 | ↓0.9 | 60.3 | ↑2.4 | 95.7 | ↑0.8 | 98.2 | ↑0.1 |
| + early $\alpha$-SVRG | 85.9 | ↑0.0 | 62.3 | ↑4.4 | 95.8 | ↑0.9 | 98.0 | ↓0.1 |

(b) **validation accuracy**

Table 11: **Early $\alpha$-SVRG on smaller classification datasets.**

Figure 11 shows the results. We can see early $\alpha$-SVRG consistently reduces training loss across different datasets. Furthermore, we observe that the validation accuracy achieved by early $\alpha$-SVRG is sometimes higher than that of standard $\alpha$-SVRG. This phenomenon is likely because disabling $\alpha$-SVRG in the later training stages allows the presence of benign gradient noise during optimization. Such noise may drive the model toward local minima that exhibit better generalization properties (Smith et al., 2020; Damian et al., 2021; Li et al., 2022).

# D  ADDITIONAL RESULTS OF OPTIMAL COEFFICIENT

We provide further results on optimal coefficient in SVRG (Equation 6) below: gradient variance reduction on other datasets with SVRG using optimal coefficient (Appendix D.1), the impact of data on optimal coefficient (Appendix D.2), and the evolution of the correlation between model gradients and snapshot gradients during training (Appendix D.3).

## D.1  SVRG WITH OPTIMAL COEFFICIENT ON OTHER DATASETS

Throughout the experiments in Section 3, we mainly study the behavior of SVRG with optimal coefficient on CIFAR-10 dataset. To show the generality of our approach for SVRG, we below monitor the optimal coefficient on other image classification datasets. We train a MLP-4 on Flowers / EuroSAT with SGD and SVRG using optimal coefficient for 15 / 25 epochs. As shown in Figure 15, SVRG with optimal coefficient can consistently reduce gradient variance and achieve a lower training loss than the baseline SGD.

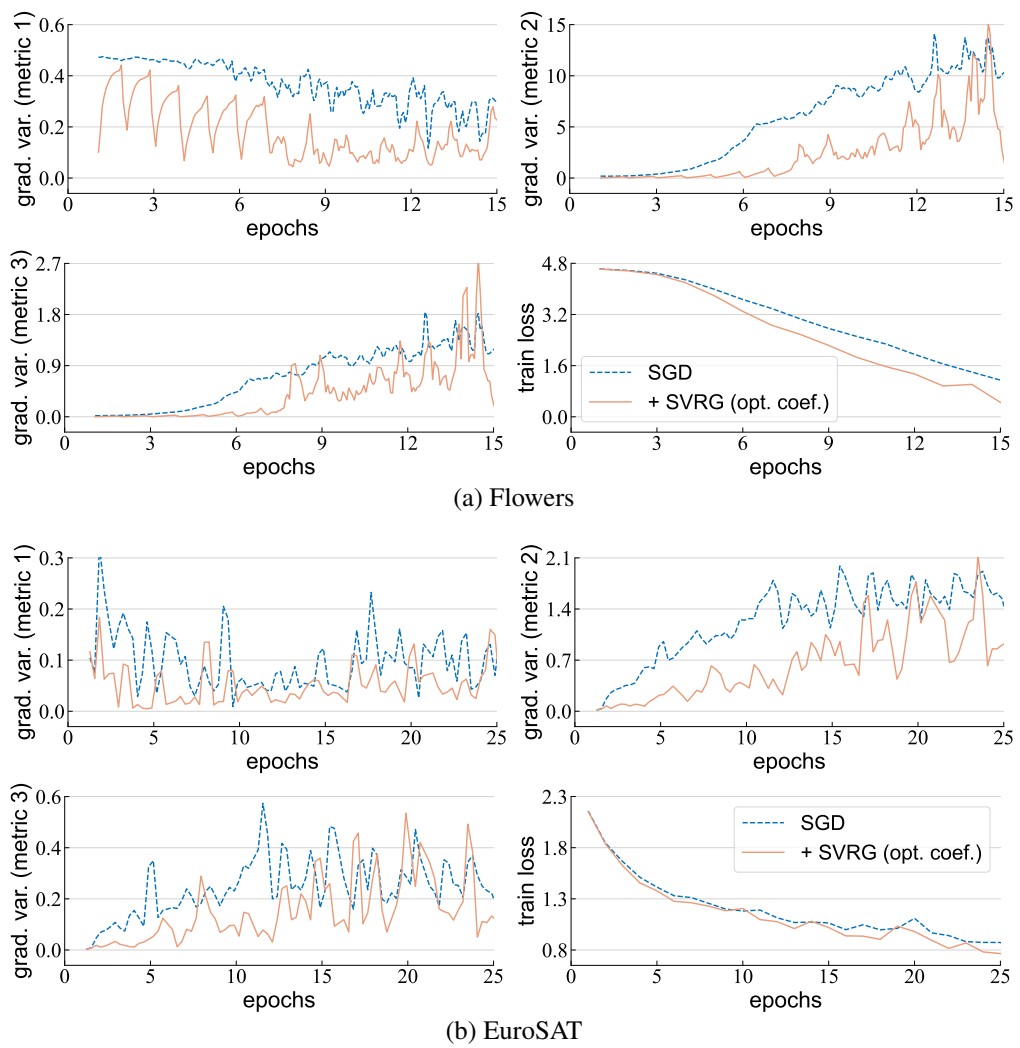

Figure 15: **SVRG with optimal coefficient on other datasets.**

## D.2  EFFECT OF DATA ON OPTIMAL COEFFICIENT

Below we conduct experiments on CIFAR-10 and CIFAR-100 to understand how the number of object classes in datasets affect optimal coefficient. Specifically, we train 1, 2, and 4-layer MLPs

(Logistic Regression, MLP-2, and MLP-4) on each of the two datasets using SGD (without SVRG) and compute the optimal coefficient (Equation 6) during the training. Figure 16 shows the results.

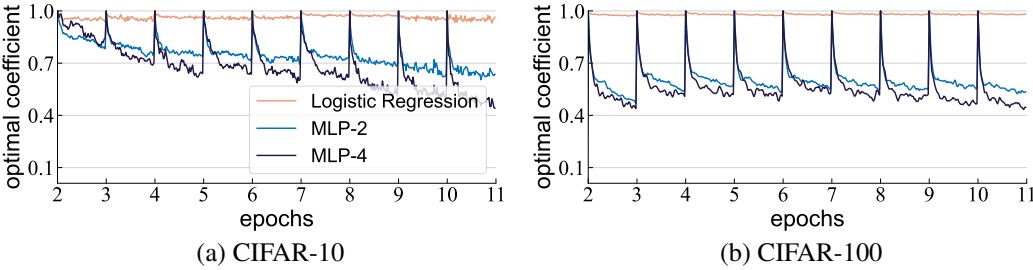

(a) CIFAR-10        (b) CIFAR-100

Figure 16: **Effects of Data on Optimal Coefficient.**

The optimal coefficient of each model trained on CIFAR-100 is lower than that on CIFAR-10. This is possibly because CIFAR-100 has $10\times$ more object classes than CIFAR-10 and therefore the correlation between the snapshot model gradient and the current model gradient is weaker. Thus, we recommend using a smaller coefficient when the training dataset includes a larger number of classes.

### D.3 CORRELATION BETWEEN MODEL GRADIENTS AND SNAPSHOT GRADIENTS

In Section 3, we find each epoch's average optimal coefficient decreases as training progresses. Here, we seek to understand whether the standard deviation ratio $\sigma(\nabla f_{\cdot,k}(\boldsymbol{\theta}^t))/\sigma(\nabla f_{\cdot,k}(\boldsymbol{\theta}^{\mathrm{past}}))$ or the correlation $\rho\left(\nabla f_{\cdot,k}(\boldsymbol{\theta}^{\mathrm{past}}), \nabla f_{\cdot,k}(\boldsymbol{\theta}^t)\right)$ in Equation 6 contributes more to this observation. We plot these two values separately in Figures 17 and 18. We can see the standard deviation ratio is relatively stable during the training, whereas the correlation decreases very much during the training.

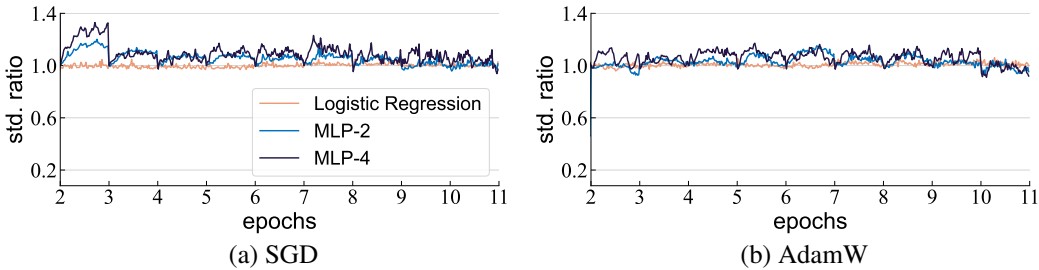

(a) SGD        (b) AdamW

Figure 17: **Standard deviation ratio.** The ratio between the standard deviations of the model gradients and the snapshot gradients oscillates around 1 but is relatively stable overall.

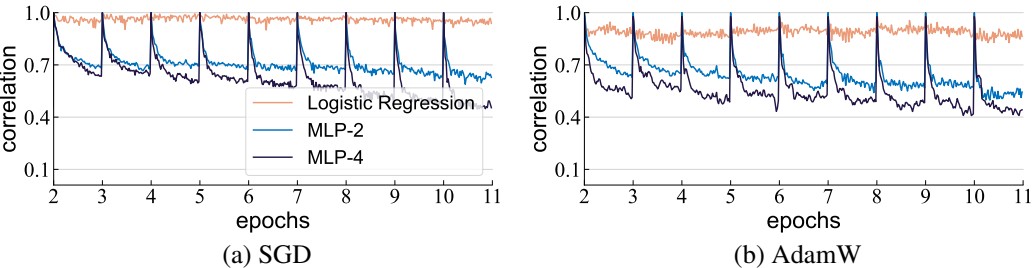

(a) SGD        (b) AdamW

Figure 18: **Correlation.** The correlation between the snapshot gradients and the model gradients decreases as the training progresses.

## E    ALTERNATIVE SCHEDULES FOR $\alpha$-SVRG

In this part, we provide the exact mathematical formulation of each schedule studied in Section 5.3.

**Notations.** We decompose the global iteration index $t$ into an epoch-wise index $s$ and an iteration index $i$ within an epoch. We also denote the total number of training epochs as $T$ and represent the number of iterations in one epoch as $M$.

**Linear schedule.** This is our default scheduling approach. The coefficient decreases linearly *across epochs and keeps as a constant within an epoch*:

$$\alpha_{\text{linear}}^t = \alpha_0(1 - \frac{s(t)}{T}), \tag{13}$$

**Other global schedules.** We also consider quadratic decay and geometric decay:

$$\alpha_{\text{quadratic}}^t = \frac{\alpha_0}{T^2}(T - s(t))^2, \tag{14}$$

$$\alpha_{\text{geometric}}^t = \alpha_0(\frac{\alpha_{\text{final}}{}^1}{\alpha_0})^{\frac{s(t)}{T}}, \tag{15}$$

**Double schedules.** In Figure 4 of Section 3, within each epoch, the coefficient starts from 1 and decreases over time. Motivated by this local behavior, we introduce three additional schedules that combine both the local and global decay: d(ouble)-linear, d-quadratic, and d-geometric. In addition to the global decay, each double schedule has another local decay for each epoch that initiates at 1 and decreases to an ending value specified by the global decay.

$$\alpha_{\text{d-linear}}^t = \alpha_{\text{linear}}^t \underbrace{(1 - \frac{i}{M})}_{\text{local decay}} + \alpha_{\text{linear}}^t \tag{16}$$

$$\alpha_{\text{d-quadratic}}^t = (1 - \alpha_{\text{quadratic}}^t) \underbrace{\frac{1}{M^2}(M - i)^2}_{\text{local decay}} + \alpha_{\text{quadratic}}^t \tag{17}$$

$$\alpha_{\text{d-geometric}}^t = \underbrace{(\alpha_{\text{geometric}}^t + \alpha_{\text{d-final}})^{\frac{i}{M}}}_{\text{local decay}} \tag{18}$$

## F  CONVERGENCE COMPARISON

We compare the training loss convergence of the AdamW baseline and $\alpha$-SVRG on ImageNet-1K (Figure 19) and small classification datasets (Figure 20). It is observed that $\alpha$-SVRG can consistently decrease training loss and deliver faster convergence.

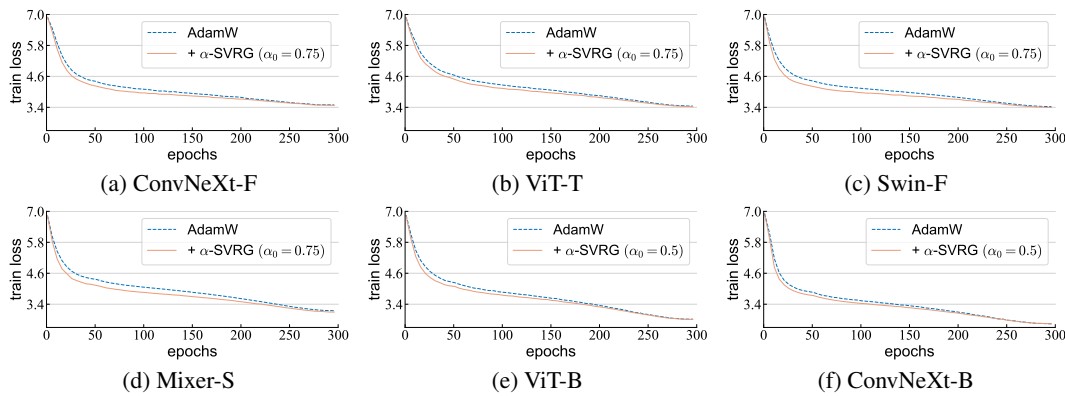

Figure 19: **Training loss with AdamW and $\alpha$-SVRG on ImageNet-1K (Table 2).**

---

[1]We set $\alpha_{\text{final}} = 0.01$ to ensure that the geometric schedule eventually decreases to a sufficiently small value. Note that $\alpha_{\text{final}}$ can not be zero, as it serves as the base of the exponent. The same rule applies to Equation 18.

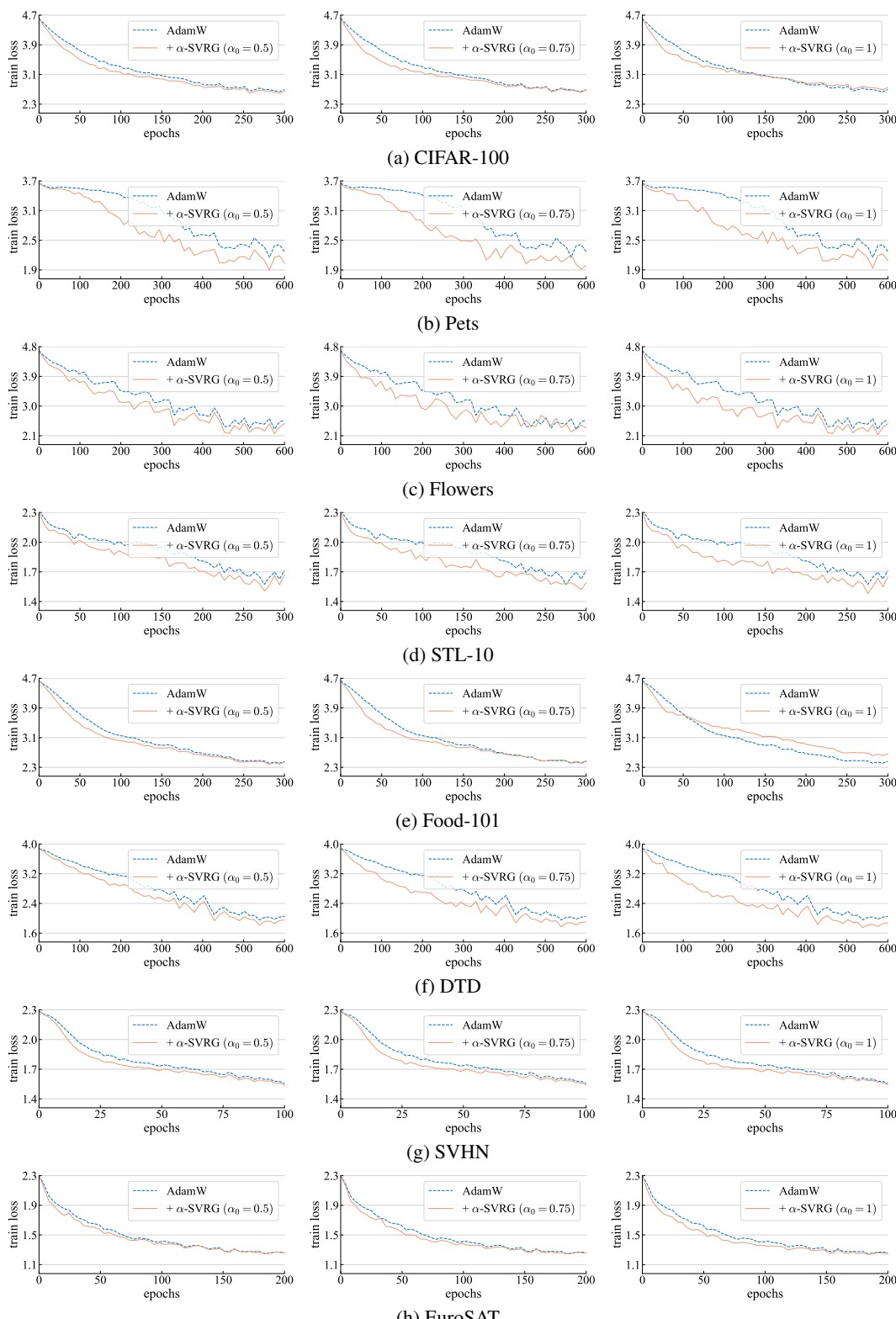

Figure 20: **Training loss with AdamW and $\alpha$-SVRG on smaller classification datasets (Table 7).**

## G PSEUDOCODE FOR $\alpha$-SVRG

For clarity, we adopt the following notations: $s$ as the epoch index, $t$ as the iteration index within each epoch, $i_t$ as the sampled index at iteration $t$, $T$ as the epoch length, $M$ as the iteration length within each epoch, $n$ as the total number of training data, and $\eta_t^s$ as the learning rate. We illustrate $\alpha$-SVRG with SGD in Algorithm 1 and with AdamW in Algorithm 2:

---

**Algorithm 1** $\alpha$-SVRG with $\mathrm{SGD}\big(\boldsymbol{\theta}_0^0, T, M, \{\{\eta_t^s\}_{t=0}^{M-1}\}_{s=0}^{T-1}, \{\{\alpha_t^s\}_{t=0}^{M-1}\}_{s=0}^{T-1}, \lambda\big)$

---

**Input:** initialized model parameters $\boldsymbol{\theta}_0^0$, epoch length $T$, iteration length within each epoch $M$, learning rates $\{\{\eta_t^s\}_{t=0}^{M-1}\}_{s=0}^{T-1}$, scheduled coefficients $\{\{\alpha_t^s\}_{t=0}^{M-1}\}_{s=0}^{T-1}$, weight decay $\lambda$

$\boldsymbol{\theta}_{\mathrm{past}}^0 = \boldsymbol{\theta}_0^0$

**for** $s = 0$ **to** $T - 1$ **do**

    $\nabla f(\boldsymbol{\theta}_{\mathrm{past}}^s) = \frac{1}{n}\sum_{i=1}^n \nabla f_i(\boldsymbol{\theta}_{\mathrm{past}}^s)$

    **for** $t = 0$ **to** $M - 1$ **do**

        $i_t \overset{\mathrm{iid}}{\sim} \mathrm{Uniform}\{1, \cdots, n\}$

        $\boldsymbol{g}_t^s = \nabla f_{i_t}(\boldsymbol{\theta}_t^s) - \alpha_t^s\big(\nabla f_{i_t}(\boldsymbol{\theta}_{\mathrm{past}}^s) - \nabla f(\boldsymbol{\theta}_{\mathrm{past}}^s)\big)$

        $\boldsymbol{\theta}_{t+1}^s = \boldsymbol{\theta}_t^s - \eta_t^s\boldsymbol{g}_t^s - \lambda\boldsymbol{\theta}_t^s$

    **end for**

    $\boldsymbol{\theta}_{\mathrm{past}}^{s+1} = \boldsymbol{\theta}_M^s$

**end for**

**Output:** final model $\boldsymbol{\theta}_M^{T-1}$.

---

---

**Algorithm 2** $\alpha$-SVRG with $\mathrm{AdamW}\big(\boldsymbol{\theta}_0^0, T, M, \{\{\eta_t^s\}_{t=0}^{M-1}\}_{s=0}^{T-1}, \{\{\alpha_t^s\}_{t=0}^{M-1}\}_{s=0}^{T-1}, \beta_1, \beta_2, \lambda\big)$

---

**Input:** initialized model parameters $\boldsymbol{\theta}_0^0$, epoch length $T$, iteration length within each epoch $M$, learning rates $\{\{\eta_t^s\}_{t=0}^{M-1}\}_{s=0}^{T-1}$, scheduled coefficients $\{\{\alpha_t^s\}_{t=0}^{M-1}\}_{s=0}^{T-1}$, momentums $\beta_1, \beta_2$, weight decay $\lambda$

$\boldsymbol{\theta}_{\mathrm{past}}^0 = \boldsymbol{\theta}_0^0$

$\boldsymbol{m} = \boldsymbol{v} = 0$

**for** $s = 0$ **to** $T - 1$ **do**

    $\nabla f(\boldsymbol{\theta}_{\mathrm{past}}^s) = \frac{1}{n}\sum_{i=1}^n \nabla f_i(\boldsymbol{\theta}_{\mathrm{past}}^s)$

    **for** $t = 0$ **to** $M - 1$ **do**

        $i_t \overset{\mathrm{iid}}{\sim} \mathrm{Uniform}\{1, \cdots, n\}$

        $\boldsymbol{g}_t^s = \nabla f_{i_t}(\boldsymbol{\theta}_t^s) - \alpha_t^s\big(\nabla f_{i_t}(\boldsymbol{\theta}_{\mathrm{past}}^s) - \nabla f(\boldsymbol{\theta}_{\mathrm{past}}^s)\big)$

        $\boldsymbol{m} = \beta_1\boldsymbol{m} + (1 - \beta_1)\boldsymbol{g}_t^s$

        $\boldsymbol{v} = \beta_2\boldsymbol{v} + (1 - \beta_2)(\boldsymbol{g}_t^s)^2$

        $\boldsymbol{\theta}_{t+1}^s = \boldsymbol{\theta}_t^s - \eta_t^s\frac{\boldsymbol{m}}{\sqrt{\boldsymbol{v}}+\epsilon} - \lambda\boldsymbol{\theta}_t^s$

    **end for**

    $\boldsymbol{\theta}_{\mathrm{past}}^{s+1} = \boldsymbol{\theta}_M^s$

**end for**

**Output:** final model $\boldsymbol{\theta}_M^{T-1}$.

---

