# OpenReview forum: "A Coefficient Makes SVRG Effective"
_ICLR.cc/2025/Conference — ICLR 2025 Poster_

### Official Review · Reviewer_4HSX · 2024-10-24

**Soundness:** 3
**Presentation:** 3
**Contribution:** 2
**Rating:** 5
**Confidence:** 4

**Summary:**

The paper studies a modification of the variance-reduced method SVRG. The authors attempted to fix the issue of the applicability of variance-reduced methods in deep learning which was noticed by Defazio & Bottou, 2019. The modification consists of adding a scaler to the variance term in the update of SVRG. The authors explain this change by considering an example of the reconstruction of one random variable X from another variable Y correlated to X. The authors provide the convergence guarantees for the proposed modification of SVRG and demonstrate its effectiveness for several DL benchmarks.

**Strengths:**

- The paper is well-written and easy to follow.

- There is a clear intuition as to why there should be a scaler smaller than 1 in the update of SVRG from the example in (2). The authors support this change by computing theoretical values of the scaler $\alpha$ for logistic regression and MLPs. It turns out that the values of  $\alpha$ drop to the end of the training for MLP and stay stable around $1$ for logistic regression. This empirical evaluation separates convex logistic regression (where SVRG is known to perform well) and non-convex MLP (where SVRG doesn't perform well).

- The proposed $\alpha$-SVRG method was tested on several DL workloads demonstrating its potential improvement over other baselines like SGD, SGD+SVRG, and AdamW.

- The authors provide potentially useful $\alpha$-schedulers that gradually decrease the values of $\alpha$. This is explained by the fact that vanilla SVRG can reduce the variance at the beginning of the training (where the stochastic gradients are more aligned) and fails in the end.

**Weaknesses:**

- SVRG-type methods do not suit the DL framework well not only because of the failure to effectively reduce the variance but also because of the necessity of full gradient computation from time to time. From this perspective, methods Zero-SARAH are more interesting to explore.

- The convergence of $\alpha$-SVRG is presented under the assumption of bounded gradients $\|\nabla f_i(x)\|\le \sigma$. In my opinion, this assumption is strong and doesn't allow to show the theoretical benefit of using $\alpha$-SVRG instead of vanilla SVRG.

- The authors claim that $\alpha$-SVRG decreases the variance. I don't understand why this is the case. For example, based on Figure 5 $\alpha$-SVRG can reduce the variance only in the beginning (where vanilla SVRG can do that as well) but afterward, the variance of $\alpha$-SVRG is similar to SGD which doesn't have any variance reduction mechanism inside. Moreover, the fluctuations of empirical variance of $\alpha$-SVRG are higher than that of SGD. Therefore, I believe it would make sense to average the variance across several random seeds to see the statistical error (it might happen that in the beginning $\alpha$-SVRG was a bit lucky). This would be also beneficial for the train loss dynamics to understand how significant the improvement. For example, for some workloads, the improvement in Table 3 is in the second digit after the dot which most likely is within the standard deviation which I believe cannot be used as significant (the results in table 8 seem to support my comment).

- The comment above also relates to the claims of the form "...can reduce gradient variance significantly..." (see line 353 and other lines like this). First, since the results of one run are presented it is not possible to measure the significance (it might be that the differences in training dynamics of the algorithm and its $\alpha$-SVRG version are within standard deviation). Second, how is "significance" measured for the proposed 3 variance metrics? Can the change 0.1 be considered large? What is the threshold for being "significant"?

- The results in Figure 10 are controversial to the claims of the paper. For small batch sizes you expect $\alpha$-SVRG to perform better since the variance should be larger. However, AdamW without any variance reduction mechanism handles the variance better than.

- The authors use various deep learning techniques like warmup in the training and the improvement might come from a weird combination of all of those techniques with $\alpha$-SVRG update. In my view, it is important to understand if $\alpha$-SVRG procedure alone indeed helps to reduce variance.

To summarize, in my view, the authors didn't provide enough empirical evidence to showcase the usefulness of $\alpha$-SVRG even though the idea of adding a scaler is intuitively logical.  The improvement is typically marginal (within standard deviation) and comes with a significant increase in memory usage and full gradient computation.

**Questions:**

- Could you please elaborate how $\alpha$-SVRG update is plugged in AdamW? Do you perform $\alpha$-SVRG update on the stochastic gradient which is then used in AdamW? Or do you perform $\alpha$-SVRG trick for the updates of AdamW? In general, this would be beneficial to add a pseudocode at least in the appendix how to combine the proposed $\alpha$-SVRG update with any optimizer.

- I don't understand the connection of $\alpha$-SVRG and preconditioning. In (38) the authors define a matrix A. Could you provide the exact form for A? I believe the SVRG update cannot be presented as a linear mapping and therefore this connection seems to be misleading.

---

> ### Author Response · Authors · 2024-11-23
> **Rebuttal by Authors**
>
> We are encouraged that you find our work tries to fix the failure of variance reduction methods in deep learning. We address your concerns and questions below:
>
> > W1: SVRG-type methods do not suit the DL framework well not only because of the failure to effectively reduce the variance but also because of the necessity of full gradient computation from time to time. From this perspective, methods Zero-SARAH are more interesting to explore.
>
> We agree SVRG-type methods require periodic full gradient computation. **Nevertheless, it is possible to compute the full gradient by first dividing the data into mini-batches and then accumulating the gradients through the mini-batches.** It does increase the computation cost. Our implementation has reduced this to 1$\times$ by storing the individual snapshot mini-batch gradient $\nabla f_i(\theta^\text{past})$ in the memory when calculating the snapshot full batch gradient $\nabla f(\theta^\text{past})$ at the beginning of each epoch.
>
>
> **Below we demonstrate we can reduce the additional computation cost of $\alpha$-SVRG compared with AdamW by applying $\alpha$-SVRG only in the early phase.** Specifically, we only apply $\alpha$-SVRG ($\alpha_0=0.75$) **during the initial 10% of the training** and add transition epoch for the coefficient schedule, where the coefficient decreases from its original value to 0. Empirically, we find this transition epoch crucial for maintaining the stability of momentum in the base optimizer. The results are shown below.
>
> Train loss
> |  | CIFAR-100 | Pets | Flowers | STL-10 | Food-101 | DTD | SVHN | EuroSAT |
> |---|---|---|---|---|---|---|---|---|
> | AdamW | 2.659 | 2.203 | 2.400 | 1.641 | 2.451 | 1.980 | 1.588 | 1.247 |
> | + $\alpha$-SVRG | 2.646 | 2.004 | 2.162 | 1.568 | 2.461 | 1.829 | 1.573 | 1.237 |
> | + early $\alpha$-SVRG | 2.644 | 2.190 | 2.328 | 1.616 | 2.444 | 1.918 | 1.583 | 1.240 |
>
> Validation accuracy
> |  | CIFAR-100 | Pets | Flowers | STL-10 | Food-101 | DTD | SVHN | EuroSAT |
> |---|---|---|---|---|---|---|---|---|
> | AdamW | 81.0 | 72.8 | 80.8 | 82.3 | 85.9 | 57.9 | 94.9 | 98.1 |
> | + $\alpha$-SVRG | 80.6 | 76.7 | 82.6 | 84.0 | 85.0 | 60.3 | 95.7 | 98.2 |
> | + early $\alpha$-SVRG | 81.0 | 74.6 | 83.6 | 82.8 | 85.9 | 62.3 | 95.8 | 98.0 |
>
> We can see that early $\alpha$-SVRG can achieve lower training losses than AdamW across all datasets. It also improves validation accuracy in most cases. **In this sense, applying $\alpha$-SVRG only in the early phase not only introduces little computational cost but also can reduce training loss and improve validation accuracy.** We have added this experiment in Appendix J.2 of the current draft.
>
> As you pointed out, Zero-SARAH [1] indeed eliminates the full gradient computation. **Nevertheless, it requires the storage of all gradients w.r.t each data point and this could be a very large memory cost for most deep learning problems.** For example, ResNet-18 has 11M parameters and if we store its gradients w.r.t each of 50K training data points in CIFAR-10, the memory cost is about 50K $\times$ 44.7MB $\approx$ 2181GB.
>
> In general, there is a trade-off between the memory cost and the additional computation cost for variance reduction methods. It is up to practitioners to choose the method that is most suitable for their specific task and hardware. **We have added a discussion of this trade-off in our related work section (line 514) and cited relevant works including Zero-SARAH.**
>
>
> [1] Li et al. ZeroSARAH: Efficient nonconvex finite-sum optimization with zero full gradient computation. arXiv'21.
>
> &nbsp;
> ---
> > W2: The convergence of $\alpha$-SVRG is presented under the assumption of bounded gradients . In my opinion, this assumption is strong and doesn't allow to show the theoretical benefit of using alpha-SVRG instead of vanilla SVRG.
>
> **Our work starts from a more empirical perspective and the convergence analysis of $\alpha$-SVRG in Appendix G is present for completeness.** We do believe there is room to further improve the convergence analysis of $\alpha$-SVRG. For example, maybe we can dynamically adjust the coefficient $\alpha$ based on the past optimization trajectory. This might require a new theoretical tool to better analyze $\alpha$-SVRG and it would be an interesting direction for future work to explore.

---

> > ### Author Response · Authors · 2024-11-23
> > **Rebuttal by Authors**
> >
> > > W3: The authors claim that alpha-SVRG decreases the variance. I don't understand why this is the case. For example, based on Figure 5 $\alpha$-SVRG can reduce the variance only in the beginning (where vanilla SVRG can do that as well) but afterward, the variance of $\alpha$-SVRG is similar to SGD which doesn't have any variance reduction mechanism inside. Moreover, the fluctuations of empirical variance of $\alpha$-SVRG are higher than that of SGD. Therefore, I believe it would make sense to average the variance across several random seeds to see the statistical error (it might happen that in the beginning $\alpha$-SVRG was a bit lucky). This would be also beneficial for the train loss dynamics to understand how significant the improvement. For example, for some workloads, the improvement in Table 3 is in the second digit after the dot which most likely is within the standard deviation which I believe cannot be used as significant (the results in table 8 seem to support my comment).
> >
> > We agree with you that the variance of $\alpha$-SVRG is not always lower than SGD. **We only observe that $\alpha$-SVRG (and standard SVRG) can reduce the variance in the early phase.** To address your concern for Figure 5, we report the average empirical gradient variance, its standard deviation, and the mean difference (between $\alpha$-SVRG and SGD) over the standard deviation of $\alpha$-SVRG ($\Delta$ mean / std) across 3 random seeds for the first 5 epochs below:
> >
> > | | metric 1 | metric 2 | metric 3 | final train loss
> > |---|---|---|---|---
> > | SGD | 0.328 ± 0.011| 1.901 ± 0.028| 0.106 ± 0.008| 0.187 ± 0.004
> > | SVRG (opt. coef.) | 0.217 ± 0.010 | 1.104 ± 0.083 | 0.061 ± 0.006 | 0.154 ± 0.013
> > | $\Delta$ mean / std | 10.9 | 9.6 | 7.1 | 2.5
> >
> > You can see that the empirical gradient variance of $\alpha$-SVRG in the early phase is lower than SGD. Moreover, the mean difference is much larger than the standard deviation of $\alpha$-SVRG. This is also the case for final training loss. **Therefore, our observation of gradient variance reduction during the early phase and train loss reduction is statistically valid.**
> >
> > **For table 8, we would like to clarify that the mean difference of training loss between $\alpha$-SVRG and baseline is always more than 1 standard deviation of $\alpha$-SVRG.** We paste the results below for your reference. You can see that $\alpha$-SVRG can consistently decrease the training loss across different datasets.
> >
> > |  | CIFAR-100 | Pets | Flowers | STL-10 | Food-101 | DTD | SVHN | EuroSAT |
> > |---|---|---|---|---|---|---|---|---|
> > | AdamW | 2.645 ± 0.013 | 2.326 ± 0.088 | 2.436 ± 0.038 | 1.660 ± 0.017 | 2.478 ± 0.021 | 2.072 ± 0.066 | 1.583 ± 0.005 | 1.259 ± 0.017 |
> > | + $\alpha$-SVRG | 2.606 ± 0.017 | 2.060 ± 0.071 | 2.221 ± 0.042 | 1.577 ± 0.022 | 2.426 ± 0.007 | 1.896 ± 0.075 | 1.572 ± 0.011 | 1.239 ± 0.01 |
> > | $\Delta$ mean / std | 2.3 | 3.7 | 5.1 | 3.8 | 7.4 | 2.3 | 1.0 | 1.2
> >
> > &nbsp;
> > ---
> > > W4: The comment above also relates to the claims of the form "...can reduce gradient variance significantly..." (see line 353 and other lines like this). First, since the results of one run are presented it is not possible to measure the significance (it might be that the differences in training dynamics of the algorithm and its $\alpha$-SVRG version are within standard deviation). Second, how is "significance" measured for the proposed 3 variance metrics? Can the change 0.1 be considered large? What is the threshold for being "significant"?
> >
> >
> > We agree with you that it is hard to quantify to what degree we can say significantly. **We have removed all wordings of “significantly” (or similar words) at all places.** This includes:
> > 1. Line 160 "substantially reduces the gradient variance"
> > 2. Line 326 "a pronounced gradient variance reduction"
> > 3. Line 353 (now line 348) "can reduce the gradient variance significantly"
> >
> > To further address your concern about the statistical significance of the results in Figure 8, we rerun the experiment over 3 random seeds and report the average empirical gradient variance across 3 runs for the first 10 epochs below. We can see the average empirical gradient variance of $\alpha$-SVRG is lower than SGD in the early phase.
> >
> > |  | metric 1 | metric 2 | metric 3 |
> > |---|---|---|---|
> > | SGD | 0.215 | 1.803 | 0.451 |
> > | $\alpha$-SVRG | 0.184 | 1.494 | 0.428 |

---

> ### Author Response · Authors · 2024-11-23
> **Rebuttal by Authors**
>
> > W5: The results in Figure 10 are controversial to the claims of the paper. For small batch sizes you expect alpha-SVRG to perform better since the variance should be larger. However, AdamW without any variance reduction mechanism handles the variance better than.
>
> **We wish to clarify that smaller batch sizes do lead to larger gradient variance. However, under larger gradient variance, $\alpha$-SVRG is not assumed to perform better.** Note that in Figure 10, we use the same coefficient of 0.5 (line 464) when changing batch size. As batch size decreases, the correlation among mini-batch gradients decreases. Using the same coefficient 0.5 might not be the best choice and we should instead decrease the coefficient to account for the weakening correlation. If we decrease the coefficient to 0.25 or 0.125, $\alpha$-SVRG can still decrease the training loss. We show the results below.
>
> | | 32 | 64 |
> |---|---|---|
> AdamW | 1.69 | 1.64 |
> $\alpha$-SVRG ($\alpha_0=0.5$) | 1.94 | 1.74 |
> $\alpha$-SVRG ($\alpha_0=0.25$) | 1.63 | 1.56 |
> $\alpha$-SVRG ($\alpha_0=0.125$) | 1.66 | 1.60 |
>
> &nbsp;
> ---
> > W6: The authors use various deep learning techniques like warmup in the training and the improvement might come from a weird combination of all of those techniques with $\alpha$-SVRG update. In my view, it is important to understand if $\alpha$-SVRG procedure alone indeed helps to reduce variance.
>
> Note that our experiments in Section 5 are strictly controlled. **We use the same training recipe for both baseline, standard SVRG, and $\alpha$-SVRG (line 371). Therefore, any improvement over the baseline should come from $\alpha$-SVRG only.** This is also applied to all initial experiments in Sections 2, 3, and 4. Moreover, for our experiments using SGD, we did not use any warmup but a constant learning rate.
>
> To see whether $\alpha$-SVRG alone can reduce gradient variance, we remove various deep learning techniques that might potentially alter loss landscape (e.g., learning rate warmup, learning rate cosine decay, mixup-cutmix, label smoothing) in our training recipe one by one. The experiments are conducted on CIFAR-100 and STL-10 and the results are reported below. We can see that even without any of these techniques, $\alpha$-SVRG can still reduce training loss. **This confirms that the improvement comes from $\alpha$-SVRG alone.**
>
> STL-10
> |  | no warmup | constant lr | mixup-cutmix | label smoothing |
> |---|---|---|---|---|
> AdamW | 1.703 | 1.731 | 0.843 | 1.540 |
> $\alpha$-SVRG ($\alpha_0=0.5$) | 1.604 | 1.720 | **0.788** | 1.437 |
> $\alpha$-SVRG ($\alpha_0=0.75$) | **1.582** | **1.703** | 0.793 | **1.397** |
> $\alpha$-SVRG ($\alpha_0=1$) | 1.588 | 1.728 | 0.836 | 1.445 |
>
> CIFAR-100
> |  | no warmup | constant lr | mixup-cutmix | label smoothing |
> |---|---|---|---|---|
> AdamW | 2.803 | 2.989 | 1.349 | 2.344 |
> $\alpha$-SVRG ($\alpha_0=0.5$) | 2.749 | 2.953 | **1.320** | **2.271** |
> $\alpha$-SVRG ($\alpha_0=0.75$) | **2.727** | **2.941** | 1.334 | 2.353 |
> $\alpha$-SVRG ($\alpha_0=1$) | 2.783 | 2.980 | 1.356 | 2.427 |
>
> &nbsp;
> ---
> > Q1: Could you please elaborate how $\alpha$-SVRG update is plugged in AdamW? Do you perform $\alpha$-SVRG update on the stochastic gradient which is then used in AdamW? Or do you perform $\alpha$-SVRG trick for the updates of AdamW? In general, this would be beneficial to add a pseudocode at least in the appendix how to combine the proposed $\alpha$-SVRG update with any optimizer.
>
> We indeed did not make it clear in the original draft. When combining $\alpha$-SVRG with AdamW, we input the variance reduced gradient $g^t_i$ (the output of $\alpha$-SVRG) into the update rule of AdamW. This practice follows other work integrating SVRG into alternative optimizer algorithms [1, 2]. We have also added the pseudo-code for $\alpha$-SVRG with AdamW in Algorithm 2 of Appendix G.
>
> [1] Dubois-Taine et al. SVRG meets adagrad: Painless variance reduction. Machine Learning'22.\
> [2] Wang et al. Divergence results and convergence of a variance reduced version of adam. arXiv'22.
>
> &nbsp;
> ---
> > Q2: I don't understand the connection of $\alpha$-SVRG and preconditioning. In (38) the authors define a matrix A. Could you provide the exact form for A? I believe the $\alpha$-SVRG update cannot be presented as a linear mapping and therefore this connection seems to be misleading.
>
> After careful review, we find it is not possible to derive $A$ in a closed form. We have removed the paragraph discussing this connection from Appendix G.
>
> &nbsp;
> ---
> We thank you for your valuable feedback and see resolving your questions and concerns as a great improvement to our paper. If you have any further questions or concerns, we are very happy to answer.

---

> ### Comment · Reviewer_4HSX · 2024-11-23
> **Response to authors**
>
> I would like to thank the authors for such a significant effort in addressing the concerns. The provided empirical results do help to understand the performance of $\alpha$-SVRG better. I still believe this method is far from being practical for the following reasons:
> - I appreciate the authors idea to use $\alpha$-SVRG in the early stages only and gradually decrease $\alpha$ to 0. However, this means that there is additional tuning: $(i)$ how long should we use $\alpha$-SVRG, how to detect that $\alpha$-SVRG stops reducing the variance? $(ii)$ How to decrease $\alpha$, is the linear decrease good enough or should we use cosine/exponential decrease/something else?
> - In several applications like training large language models (typically pertaining is performed by doing only a few passes over the data) or training RL models (here we don't have access to the full gradient at all). In my opinion, computing the full gradient is a significant limitation of $\alpha$-SVRG in modern applications where both model and data sizes are huge. Therefore, the authors should have proposed a more practical algorithm where the full gradient can be somehow approximated.
>
> Nonetheless, based o the provided set of experiments I increase my score.

---

> ### Author Response · Authors · 2024-11-24
> **Further Response by Authors**
>
> Thank you for your quick response to our rebuttal! We address your further concerns below:
>
> > Q1: I appreciate the authors idea to use $\alpha$-SVRG in the early stages only and gradually decrease $\alpha$ to 0. However, this means that there is additional tuning: $(i)$ how long should we use $\alpha$-SVRG, how to detect that $\alpha$-SVRG stops reducing the variance? $(ii)$ How to decrease $\alpha$, is the linear decrease good enough or should we use cosine/exponential decrease/something else?
>
> We are currently preparing an updated response for Q1, and please refer to the newer version once it is posted. Thank you and sorry for the confusion.
>
>
> &nbsp;
> ---
>
> > Q2: In several applications like training large language models (typically performed by doing only a few passes over the data) or training RL models (here we don't have access to the full gradient at all), computing the full gradient is a significant limitation of $\alpha$-SVRG. In my opinion, the authors should have proposed a more practical algorithm where the full gradient can be somehow approximated.
>
> We agree with you that computing the full gradient is indeed not possible in some deep learning problems. Below we discuss three potential ways to address this limitation.
>
> 1. We can use the idea of mega-batch [1, 2, 3] in SVRG-type methods. They typically use 10-32 $\times$ mini-batch samples to estimate the full gradient. In this way, we can still form the variance reduced gradient $g^t_i$ by replacing the full gradient with its mega-batch approximation and then updating the model with the variance reduced gradient.
>
> 2. There is also recent work from last week ([https://arxiv.org/abs/2411.10438](https://arxiv.org/abs/2411.10438)) that adapts the variance reduction method to train Large Language Models. They eliminate the need to compute the full gradient by only using mini-batch gradient $\nabla f_i(\theta^t)$ and snapshot gradient $\nabla f_i(\theta^{\text{past},t})$ to form the variance reduction term. Moreover, similar to us, they also employ a coefficient to control the variance reduction strength.
>
> 3. We can also use the EMA of the past snapshot gradients to approximate the full gradient. Specifically, we first form the snapshot model as the EMA of the past model parameters
> $\theta^{\text{past},t+1} = \beta_1 \theta^{\text{past},t} + (1 - \beta_1) \theta^{t}$,
> and the full batch gradient can then be approximated by the EMA of the past snapshot mini-batch gradients:
> $m^{t+1} = \beta_2 m^t + (1 - \beta_2) \nabla f_i(\theta^{\text{past},t})$. The variance reduced gradient is still constructed as before:
> $g^t_i = \nabla f_i(\theta^t) - \alpha^t (f_i(\theta^{\text{past},t}) - m^t)$. We show the results using $\alpha$-SVRG with EMA ($\beta_1=0.9999$, $\beta_2=0.9$) on various datasets. We can see that $\alpha$-SVRG (EMA) can achieve competitive results, compared with standard $\alpha$-SVRG. More importantly, $\alpha$-SVRG (EMA) consistently outperforms AdamW across different datasets.
>
> Train loss
> |  | ImageNet-1K |CIFAR-10 |CIFAR-100 | Pets | Flowers | STL-10 | Food-101 | DTD |
> |---|---|---|---|---|---|---|---|---|
> | AdamW | 3.487 | 1.407 | 2.659 | 2.203 | 2.400 | 1.641 | 2.451 | 1.980 |
> | + $\alpha$-SVRG | 3.471 | **1.389** | 2.646 | **2.004** | **2.162** | **1.568** | 2.461 | **1.829** |
> | + $\alpha$-SVRG (EMA) | **3.455** | 1.391 | **2.583** | 2.161 | 2.252 | 1.596 | **2.395** | 1.963 |
>
> Validation accuracy
> |  | ImageNet-1K | CIFAR-10 | CIFAR-100 | Pets | Flowers | STL-10 | Food-101 | DTD |
> |---|---|---|---|---|---|---|---|---|
> | AdamW | 76.0 | 96.7 |81.0 | 72.8 | 80.8 | 82.3 | 85.9 | 57.9 |
> | + $\alpha$-SVRG | 76.3 | 96.7 | 80.6 | **76.7** | **82.6** | 84.0 | 85.0 | **60.3** |
> | + $\alpha$-SVRG (EMA) | **76.5** | 96.7 | **82.0** | 74.8 | 82.3 | **84.1** | **86.4** | 59.5 |
>
> Overall, we agree that our current setup for $\alpha$-SVRG might not be well-suited for all deep learning problems, such as training Large Language Models. **Nevertheless, our work has demonstrated the potential of SVRG-type methods in training real-world neural networks.** It would be of great value for future work to improve our method further and make it widely applicable in deep learning.
>
> [1] Frostig et al. Competing with the empirical risk minimizer in a single pass. COLT'15.\
> [2] Lei et al. Non-convex finite-sum optimization via scsg methods. NeurIPS'17.\
> [3] Papini et al. Stochastic variance-reduced policy gradient. ICML'18.
>
> &nbsp;
> ---
>
> Again, we thank you for your constructive comments and they are always improving our paper! If you have any further questions or concerns, we are very happy to answer.

---

> ### Author Response · Authors · 2024-11-28
> **Further Response by Authors**
>
> Please note that we have updated our previous response for Q2 above and posted a new response for Q1 here.
>
> &nbsp;
> ---
>
> > Q1 (1): I appreciate the authors idea to use $\alpha$-SVRG in the early stages only and gradually decrease $\alpha$ to 0. However, this means that there is additional tuning: $(i)$ how long should we use $\alpha$-SVRG, how to detect that $\alpha$-SVRG stops reducing the variance?
>
> We agree that how long to apply early $\alpha$-SVRG is another hyperparameter for our method. To understand this better, we vary the length of applying early $\alpha$-SVRG (using a linear schedule with $\alpha_0=0.75$ and a transition epoch) from 2.5% to 20% of the training and the results are reported below. **$\alpha$-SVRG can achieve a lower training loss compared to the baseline AdamW with as little as 5% of the total training time. The reduction in training loss becomes much more pronounced as we extend the time to apply early $\alpha$-SVRG.**
>
> | | CIFAR-100 | Pets | Flowers | STL-10 | Food-101 | DTD | SVHN | EuroSAT |
> |---|---|---|---|---|---|---|---|---|
> | AdamW | 2.659 | 2.203 | 2.400 | 1.641 | 2.451 | 1.980 | 1.591 | 1.247 |
> | + early $\alpha$-SVRG (2.5%) | 2.663 | 2.355 | 2.444 | 1.651 | 2.473 | 1.988 | 1.635 | 1.246 |
> | + early $\alpha$-SVRG (5.0%) | 2.652 | 2.222 | 2.382 | 1.636 | 2.465 | 1.946 | 1.585 | 1.240 |
> | + early $\alpha$-SVRG (10.0%) | 2.644 | 2.190 | 2.328 | 1.616 | 2.444 | 1.918 | 1.583 | 1.240 |
> | + early $\alpha$-SVRG (20.0%) | 2.641 | 2.137 | 2.285 | 1.590 | 2.463 | 1.888 | 1.579 | 1.236 |
>
> **For your concern about how to detect that $\alpha$-SVRG stops reducing the gradient variance, we can leverage all the mini-batch gradients $\nabla f_i(\theta^t)$ and variance reduced gradients $g^t_i$ in each epoch to determine the stopping point.** If the gradient variance estimated by variance reduced gradients $Var(g^t_i)$ is larger than that by mini-batch gradients $Var(\nabla f_i(\theta^t))$, we should disable $\alpha$-SVRG thereafter.
>
>
> &nbsp;
> ---
>
> > Q1 (2): How to decrease $\alpha$, is the linear decrease good enough or should we use cosine/exponential decrease/something else?
>
> There are indeed different ways to decrease $\alpha$ to 0 for early $\alpha$-SVRG. Our previous experiments only use a single epoch to decrease $\alpha$ to 0. **Below we extend the length of this transition period to multiple epochs ($\{1,2,4\}$) and compare different schedules (linear, cosine, exponential, quadratic).**
>
> CIFAR-100 (baseline: 2.659)
> |  | transition epochs = 1 | transition epochs = 2 | transition epochs = 4 |
> |---|---|---|---|
> | linear | 2.644 | **2.643** | 2.647 |
> | cosine | 2.646 | 2.648 | 2.646 |
> | exponential | 2.650 | 2.647 | 2.645 |
> | quadratic | 2.647 | 2.646 | 2.644 |
>
> Flowers (baseline: 2.400)
> |  | transition epochs = 1 | transition epochs = 2 | transition epochs = 4 |
> |---|---|---|---|
> | linear | **2.328** | 2.339 | 2.343 |
> | cosine | 2.341 | 2.345 | 2.336 |
> | exponential | 2.356 | 2.338 | 2.378 |
> | quadratic | 2.334 | 2.327 | 2.333 |
>
> Pets (baseline: 2.203)
> |  | transition epochs = 1 | transition epochs = 2 | transition epochs = 4 |
> |---|---|---|---|
> | linear | 2.190 | **2.172** | 2.216 |
> | cosine | 2.190 | 2.174 | 2.215 |
> | exponential | 2.197 | 2.195 | 2.183 |
> | quadratic | 2.201 | 2.209 | 2.178 |
>
> STL-10 (baseline: 1.641)
> |  | transition epochs = 1 | transition epochs = 2 | transition epochs = 4 |
> |---|---|---|---|
> | linear | **1.616** | 1.634 | 1.625 |
> | cosine | 1.631 | 1.637 | 1.627 |
> | exponential | 1.635 | 1.631 | 1.634 |
> | quadratic | 1.627 | 1.633 | 1.635 |
>
> **The results show that early $\alpha$-SVRG can achieve a lower training loss than the baseline AdamW regardless of the schedule.** Among all the schedules, the linear schedule decreases the training loss the most and the optimal length of the transition epoch is usually 1 or 2. Therefore, we recommend using the linear schedule and transition epoch length 1 in practice.
>
> &nbsp;
> ---
>
> We hope our above responses can address your questions. Please let us know if you have any further questions or concerns!

---

> > ### Author Response · Authors · 2024-11-30
> > **Kind reminder to review our further response**
> >
> > Dear Reviewer 4HSX,
> >
> > We hope this message finds you well. We are writing to kindly ask you to review our response to your further questions about $\alpha$-SVRG. Please let us know if you have any further questions and we are happy to answer them.
> >
> > Thank you for your time and consideration!
> >
> > Best,\
> > Authors

---

> > > ### Comment · Reviewer_4HSX · 2024-12-02
> > > **Response to authors**
> > >
> > > I appreciate the authors' effort to address my concerns. Most of them were addressed, and empirical results demonstrate the potential of variance-reduced methods. However, in my view, there is a lack of mathematical foundation and intuition behind the method. The proposed intuition relies on eq. (2) and (3) which involve computing the correlation between current stochastic gradient and past gradients. This intuition surely works for convex functions, but the correlation in the highly non-convex training of neural networks seems to be far from just linear as eq. (2) suggests. This is supported by the fact that the authors have to use different schedulers to set the parameter $\alpha$ in the experiments. As a result, this introduces one more level in the tuning process on top of the learning rate parameter, weight decay parameter, learning rate scheduler, etc. Therefore, in the current state, I am not convinced that the proposed algorithm is good enough to be used in practice. Therefore, I keep my score of 5 (which was already adjusted because of the experiments that the authors provided during the rebuttals).

---

> ### Author Response · Authors · 2024-12-03
> **Further response by authors**
>
> Thank you for your further reply!
>
> Regarding the intuition behind our setup, we acknowledge that gradient correlations in non-convex models may not strictly follow the linear relationships in Equation (2). **Nevertheless, the original SVRG has already assumed a linear structure: using $\nabla f_i(\theta^{\text{past}}) - \nabla f(\theta^{\text{past}})$ to reduce variance in $f_i(\theta^t)$.** Our proposed $\alpha$-SVRG only adds a coefficient $\alpha$ to this linear structure and aims to address other shortcomings of SVRG.
>
> We also acknowledge that our $\alpha$-SVRG introduces one more hyperparameter $\alpha_0$. **However, we view it more as a missing hyperparameter that should have been there for the original SVRG, for effective variance reduction.** We also showed that setting $\alpha_0$ at different values between 0 and 1 can generally improve or at least not hurt the performance (Table 7 in Appendix B), although the best value may vary.
>
> Regarding your comment “the authors have to use different schedulers to set the parameter $\alpha$ in the experiments”, **we wish to clarify that our main results (Table 2 and Table 3) always use a linear scheduler, and the search of the initial coefficients $\alpha_0$ is only done under the linear scheduler.** Our experiments (Table 4) also show that $\alpha$-SVRG with a linear schedule (our default recommendation) can already achieve lower training loss than baseline AdamW.
>
> Thank you again for reviewing our paper and engaging in discussions. We are encouraged that our rebuttal addressed most of your concerns, and we appreciate the rating increase from 3 to 5. We hope these final clarifications can help address your remaining concerns.

---

### Official Review · Reviewer_9C3X · 2024-11-02

**Soundness:** 3
**Presentation:** 4
**Contribution:** 2
**Rating:** 6
**Confidence:** 4

**Summary:**

This paper proposed the $\alpha$-SVGR algorithm, which introduces a linearly decaying coefficient to adjust the scale of the correction term in SVRG. The proposed algorithm is inspired by the observation that the original SVRG does not effectively reduce the variance in the late training phase. To show the effectiveness of $\alpha$-SVRG, the authors conducted extensive experiments across various datasets and model architectures.

**Strengths:**

1. This paper is well-written with clear logic. It first identifies the problem with the original SVRG by monitoring the gradient variance during training. Then, it addresses the identified issue by introducing an extra coefficient that adjusts the scale of the SVRG correction.

2. The evaluation is comprehensive. The authors not only conducted extensive experiments across various datasets and model architectures but also explored other possible schedules for the introduced hyperparameter $\alpha$. However, I still have some concerns about the effectiveness and extra cost of $\alpha$-SVRG (see weaknesses).

**Weaknesses:**

**Major**
1. The effectiveness of $\alpha$-SVRG: In Table 2, compared with the baseline $\alpha$-SVRG's improvement in training loss and validation accuracy is marginal.

2. The computation and memory cost of $\alpha$-SVRG: $\alpha$-SVRG requires additional computations for the full gradient $\nabla f(\theta^{\text{past}})$ and the mini-batch gradient at $\theta^{\text{past}}$, $\nabla f_i(\theta^{\text{past}})$. Additionally, it necessitates maintaining $\theta^{\text{past}}$ and $\nabla f(\theta^{\text{past}})$ in memory. The paper fails to address or discuss these significant costs.


In summary, while the paper offers a good starting point for investigating why SVRG underperforms in deep learning, the enhancements to the original SVRG model, represented by $\alpha$-SVRG, do not sufficiently justify the substantial additional costs and provide only limited benefits compared to standard SGD or AdamW.


**Minor**:
1. The authors are advised to include the pseudo-code for AdamW + $\alpha$-SVRG rather than merely describing its use of AdamW as the base optimizer.

2. Typos in eqs (7)-(9).

**Questions:**

In Figure 5, the gradient variance of $\alpha$-SVRG is not lower than SGD in the late phase in terms of any metric. Does the improvement of $\alpha$-SVRG in training loss, if any, come from the reduced variance in the early phase? If this is the case, would it be more effective to employ $\alpha$-SVRG only during the initial phase and switch to standard AdamW or SGD for the later stages?

To further elaborate my question, the role of gradient noise in helping generalization is widely discussed in the literature (e.g., [1]-[5]). In the late phase of training, we even need to encourage more helpful, benign noise that drives the model parameter to better generalizing minima. From this perspective, why not altogether remove the correction term in SVRG in the late phase?

[1] Samuel Smith, Erich Elsen, and Soham De. On the generalization benefit of noise in stochastic gradient descent. ICML'2020.

[2] Alex Damian, Tengyu Ma, and Jason D. Lee. Label noise SGD provably prefers flat global minimizers. NeurIPS'2021.

[3] Zhiyuan Li, Tianhao Wang, and Sanjeev Arora. What happens after SGD reaches zero loss?–a mathematical framework. ICLR'2021.

[4] Guy Blanc, Neha Gupta, Gregory Valiant, and Paul Valiant. Implicit regularization for deep neural networks driven by an ornstein-uhlenbeck like process. COLT'2020.

[5] Xinran Gu, Kaifeng Lyu, Longbo Huang, Sanjeev Arora. Why (and When) does Local SGD Generalize Better than SGD? ICLR'2023.

---

> ### Author Response · Authors · 2024-11-23
> **Rebuttal by Authors**
>
> We sincerely thank you for your comment. We appreciate you find that our work has a clear logic and our experiments are very comprehensive. We address your concerns and questions below:
>
> > W1: The effectiveness of $\alpha$-SVRG: In Table 2, compared with the baseline alpha-SVRG's improvement in training loss and validation accuracy is marginal.
>
> Our results in Table 2 are on the ImageNet-1K dataset. Improving performance on this complex dataset by a large margin can be challenging. Recent model architecture papers [1]-[4] consider accuracy improvement similar to our scale on ImageNet-1k (<1.0%) to be significant enough. **Therefore, the improvements we achieve using a single coefficient in our work are quite significant.**
>
> **Additionally, our method demonstrates a greater advantage over AdamW on other image classification datasets, as shown in Table 3.** For instance, on the Pets dataset, our approach achieves approximately an 11.0% relative reduction in training loss and a 5.0% absolute improvement in validation accuracy compared to the baseline AdamW optimizer.
>
> [1] Tu et al. MaxViT: Multi-Axis Vision Transformer. ECCV'22.\
> [2] Yu et al. MetaFormer Baselines for Vision. TPAMI'22.\
> [3] Woo et al. ConvNeXt V2: Co-designing and Scaling ConvNets with Masked Autoencoders. CVPR'23.\
> [4] Liu et al. A ConvNet for the 2020s. CVPR'22.
>
> &nbsp;
> ---
> > W2: The computation and memory cost of $\alpha$-SVRG: $\alpha$-SVRG requires additional computations for the full gradient $\nabla f(\theta^\text{past})$ and the mini-batch gradient at $\theta^\text{past}$, $\nabla f_i(\theta^\text{past})$. Additionally, it necessitates maintaining $\nabla f(\theta^\text{past})$ and $\nabla f(\theta)$ in memory. The paper fails to address or discuss these significant costs.
>
>
> **Our original intention is to primarily compare $\boldsymbol{\alpha}$-SVRG to standard SVRG.** Both of them require computing the full gradient $\nabla f(\theta^\text{past})$ and the mini-batch gradient $\nabla f_i(\theta^\text{past})$ for snapshot model. Therefore, the computation cost and memory cost of $\alpha$-SVRG is the same as standard SVRG.
>
> We agree with you that computing the full gradient $\nabla f(\theta^\text{past})$ and the mini-batch gradient $\nabla f_i(\theta^\text{past})$ for the snapshot model is not negligible. **This is indeed one fundamental problem for most variance reduction methods [1]-[7].** In particular, the way that the variance reduction works requires computing another term that is correlated with the original mini-batch gradient $\nabla f_i(\theta)$. This additional term is where the computation cost comes from. We have included a discussion of this limitation in Appendix I.
>
> **To address your concern about the memory cost, we would like to clarify that storing the full gradient $\nabla f(\theta^\text{past})$ and the mini-batch gradient $\nabla f_i(\theta^\text{past})$ for the snapshot model does not require much memory.** Specifically, we need to store two gradients, each of which is the same size as the original model. For a small model like "ResNet18" with 11M parameters studied in our paper, this memory cost is about 2 $\times$ 44.7MB; for a large model like "ConvNeXt-Base" with 88.6M parameters, the memory cost is about 2 $\times$ 337.9MB. Both of them are less than 1GB, which should not be a big issue for modern GPUs.
>
>
> We will further address your concern about the additional computation cost of $\alpha$-SVRG compared with AdamW in Q1.
>
> [1] Nguyen et al. SARAH: A novel method for machine learning problems using stochastic recursive gradient. ICLR'17.\
> [2] Lei et al. Non-convex finite-sum optimization via scsg methods. NeurIPS'17.\
> [3] Fang et al. Spider: Near-optimal non-convex optimization via stochastic path-integrated differential estimator. NeurIPS'18.\
> [4] Wang et al. Spiderboost and momentum: Faster variance reduction algorithms. NeurIPS'19.\
> [5] Cutkosky et al. Momentum-Based Variance Reduction in Non-Convex SGD. NeurIPS'19.\
> [6] Li et al. ZeroSARAH: Efficient nonconvex finite-sum optimization with zero full gradient computation. arXiv'21.\
> [7] Kavis et al. Adaptive stochastic variance reduction for non-convex finite-sum minimization. NeurIPS'22.
>
> &nbsp;
> ---
> > W3: The authors are advised to include the pseudo-code for AdamW + alpha-SVRG rather than merely describing its use of AdamW as the base optimizer.
>
> Thank you for the suggestion. We have added the pseudo-code for $\alpha$-SVRG with AdamW in Algorithm 2 of Appendix G.
>
> &nbsp;
> ---
> > W4: Typos in eqs (7)-(9).
>
> Thank you for pointing this out. In our original draft, Equations 7 - 9 are indeed not very clear. We wish to clarify that $t$ is the global iteration index, which can be decomposed into the epoch index $s$ and the iteration index $i$ within the epoch. **In other words, $\boldsymbol{s}$ is a function of $\boldsymbol{t}$.** So for clarity, we should use $s(t)$ instead of $s$ in Equations 7 - 9. We have revised these equations in the current draft.

---

> > ### Author Response · Authors · 2024-11-23
> > **Rebuttal by Authors**
> >
> > > Q1: In Figure 5, the gradient variance of $\alpha$-SVRG is not lower than SGD in the late phase in terms of any metric. Does the improvement of $\alpha$-SVRG in training loss, if any, come from the reduced variance in the early phase? If this is the case, would it be more effective to employ $\alpha$-SVRG only during the initial phase and switch to standard AdamW or SGD for the later stages? \
> > To further elaborate my question, the role of gradient noise in helping generalization is widely discussed in the literature (e.g., [1]-[4]). In the late phase of training, we even need to encourage more helpful, benign noise that drives the model parameter to better generalizing minima. From this perspective, why not altogether remove the correction term in $\alpha$-SVRG in the late phase?
> >
> > **We agree with you that the improvement of $\alpha$-SVRG in training loss primarily comes from the benefit of gradient variance reduction in the early phase.** Here we provide an experiment on various image classification datasets to show that this is indeed the case. Specifically, we only apply $\alpha$-SVRG ($\alpha_0=0.75$) **during the initial 10% of the training** and add a transition epoch for the coefficient schedule, where the coefficient decreases from its original value to 0. Empirically, we find this transition epoch crucial for maintaining the stability of momentum in the base optimizer. The results are shown below.
> >
> > Train loss
> > |  | CIFAR-100 | Pets | Flowers | STL-10 | Food-101 | DTD | SVHN | EuroSAT |
> > |---|---|---|---|---|---|---|---|---|
> > | AdamW | 2.659 | 2.203 | 2.400 | 1.641 | 2.451 | 1.980 | 1.588 | 1.247 |
> > | + $\alpha$-SVRG | 2.646 | 2.004 | 2.162 | 1.568 | 2.461 | 1.829 | 1.573 | 1.237 |
> > | + early $\alpha$-SVRG | 2.644 | 2.190 | 2.328 | 1.616 | 2.444 | 1.918 | 1.583 | 1.240 |
> >
> > Validation accuracy
> > |  | CIFAR-100 | Pets | Flowers | STL-10 | Food-101 | DTD | SVHN | EuroSAT |
> > |---|---|---|---|---|---|---|---|---|
> > | AdamW | 81.0 | 72.8 | 80.8 | 82.3 | 85.9 | 57.9 | 94.9 | 98.1 |
> > | + $\alpha$-SVRG | 80.6 | 76.7 | 82.6 | 84.0 | 85.0 | 60.3 | 95.7 | 98.2 |
> > | + early $\alpha$-SVRG | 81.0 | 74.6 | 83.6 | 82.8 | 85.9 | 62.3 | 95.8 | 98.0 |
> >
> > We can see that early $\alpha$-SVRG can achieve lower training losses than AdamW across all datasets. It also improves validation accuracy in most cases. More importantly, the validation accuracy for early $\alpha$-SVRG is sometimes higher than that of standard $\alpha$-SVRG on some datasets (e.g., Flowers, Food-101, DTD, SVHN). **This also validates your point that disabling the variance reduction in the late phase can improve the generalization performance.**
> >
> > **In this sense, applying $\alpha$-SVRG only in the early phase not only introduces little computational cost but also can reduce training loss and improve validation accuracy.** We have added this experiment in Appendix J.2 of the current draft.
> >
> > &nbsp;
> > ---
> > We thank you for your valuable feedback and see resolving your questions and concerns as a great improvement to our paper. If you have any further questions or concerns, we are very happy to answer.

---

> ### Comment · Reviewer_9C3X · 2024-11-26
> **Thank you for the detailed response and I have raised my score!**
>
> Thank you for the detailed response and for the additional experiments on early $\alpha$-SVRG. I believe the key reason why both $\alpha$-SVRG and early $\alpha$-SVRG perform well lies in the complex role of noise in modern deep learning. On the one hand, noise can hinder optimization and lead to slower convergence; on the other hand, it also induces implicit regularization, helping to find better generalizing minima. As such, there is an inherent tradeoff in controlling the noise scale, and $\alpha$-SVRG strikes an effective balance by dynamically adjusting the noise. One advantage of $\alpha$-SVRG over early $\alpha$-SVRG is that the latter introduces an additional hyperparameter — the time to switch — which adds complexity to the tuning process.
>
> Overall, I appreciate the authors' efforts in rethinking classical algorithms like SVRG, which perform well theoretically in convex settings, and adapting them to work effectively in modern deep learning. I also suggest that the authors consider including a discussion on the implicit bias induced by gradient noise in the next version of the paper.

---

> > ### Author Response · Authors · 2024-11-28
> > **Thank you for the review and discussion**
> >
> > We appreciate your helpful review and discussion about early $\alpha$-SVRG! We are glad that all your concerns have been resolved. We will add the discussion for implicit bias induced by gradient noise and how it interacts with $\alpha$-SVRG in our draft.

---

### Official Review · Reviewer_Nknp · 2024-11-03

**Soundness:** 3
**Presentation:** 3
**Contribution:** 2
**Rating:** 6
**Confidence:** 4

**Summary:**

This paper observes the ineffectiveness of SVRG due to the decay of the relevance between the batch gradient and the snapshot. The paper then proposes $\alpha$-SVRG, with $\alpha$ as a conditioning multiplicator to account for the decaying relevance. The approach is shown to be effective in multiple experiments.

**Strengths:**

1. The flow of the paper is easy to follow.
2. There is adequate discussion about the experiment results.
3. The perspective of the gradient preconditioning is interesting.

**Weaknesses:**

1. The design of the scheduling of $\alpha$ may require more discussions and refinements. Specifically,
- The effect of $\alpha_0$. The following three points are a bit inconsistent as a whole: (a). From Figure 4, the optimal $\alpha_0$ seems to be almost the same for different models. (b) From Table 4 and Figure 9, it seems that $\alpha\approx0.75$ may be a better choice that $\alpha_0=0.5$ (at least for the small model that is used). (c) In the experiment that yields Table 3, however, $\alpha-0.5$ for small models and $\alpha_0=0.75$ for larger ones.
- The schedule of $\alpha$. It may be interesting to analyze different types of scheduling of $\alpha$ based on the perspective of preconditioning in Appendix G.
2. For the experiment that yields Figure 4, it would be better if the authors could conduct experiments on different datasets and try to analyze the effect of data on the optimal $\alpha$.

**Questions:**

In the schedule of Eq. (9), why do we use a constant of $10^{-2}$ instead of other constants? Can this constant also be tuned?

---

> ### Author Response · Authors · 2024-11-23
> **Rebuttal by Authors**
>
> We sincerely thank you for your comment. We appreciate you find our method $\alpha$-SVRG effective under multiple settings. We address your concerns and questions below:
>
> >W1.1: The effect of $\alpha_0$. The following three points are a bit inconsistent as a whole: (a). From Figure 4, the optimal  $\alpha_0$ seems to be almost the same for different models. (b) From Table 4 and Figure 9, it seems that $\alpha_0=0.75$ may be a better choice than  $\alpha_0=0.5$ (at least for the small model that is used). (c) In the experiment that yields Table 3, however,  $\alpha_0=0.5$ for small models and $\alpha_0=0.75$ for larger ones.
>
> We clarify the inconsistency you mentioned before:
> 1. **In Figure 4, we did not schedule the coefficient $\boldsymbol{\alpha}$ (and $\boldsymbol{\alpha}_0$). Instead, the coefficient is computed based on Equation 6 and should be the same (with a value of 1) at the beginning of each epoch.** This is because at that time the snapshot model is exactly the same as the current model and they have a perfect correlation of 1 (Equation 6).
> 2. In contrast, in our proposed method $\alpha$-SVRG, we schedule $\alpha$ during the training. **Note that $\boldsymbol{\alpha}_0$ in the schedule does not represent the coefficient at the first iteration but the average of all the coefficients during the first epoch (Equation 7 and Figure 12).** Therefore, we set it to $\alpha_0=0.75$ to account for the average value for the coefficient during the first epoch.
> 3. **For Table 3, we use $\boldsymbol{\alpha}_0=0.75$ for smaller models and $\boldsymbol{\alpha}_0=0.5$ for larger ones (see line 370).** This choice of initial coefficient $\alpha_0$ is based on the observation that a deeper model generally has a lower coefficient in Figure 4.
>
> &nbsp;
> ---
> > W1.2: The schedule of $\alpha$. It may be interesting to analyze different types of scheduling of $\alpha$ based on the perspective of preconditioning in Appendix G.
>
> We agree with you that it may be interesting to analyze different types of scheduling of $\alpha_0$ based on the perspective of preconditioning. For example, we can utilize the information (e.g., gradient norm, variance, momentum) in the past optimization trajectory to adjust $\alpha$ dynamically and possibly eliminate the need of scheduling $\alpha$ at all. Nevertheless, our work has empirically found that simple linear decay (Equation 7) is very effective. We believe it might be interesting for future works to understand how to further improve the schedule of $\alpha$.
>
> &nbsp;
> ---
> > W2: For the experiment that yields Figure 4, it would be better if the authors could conduct experiments on different datasets and try to analyze the effect of data on the optimal alpha $\alpha^*$.
>
> Thank you for the suggestion. **We have added experiments to analyze the optimal coefficient on datasets with different numbers of object classes, in particular CIFAR-10 and CIFAR-100 below.** We train three models (Logistic Regression, MLP-2, and MLP-4) on each dataset and compute the optimal coefficient during the training. The results are available [here](https://drive.google.com/file/d/167o7ayC5TNRZsq9nvioBRBsGRYL8mGOY/view?usp=share_link). We observe that the optimal coefficient is smaller for CIFAR-100 than CIFAR-10. This is possibly because CIFAR-100 has 10x object classes than CIFAR-10, making the correlation between the snapshot model gradient $\nabla f_i(\theta^\text{past})$ and the current model gradient $\nabla f_i(\theta^t)$ lower. We have incorporated this experiment in Appendix K.1 of the current draft.
>
>
>
> &nbsp;
> ---
> > Q1: In the schedule of Eq. (9), why do we use a constant of 0.01 instead of other constants? Can this constant also be tuned?
>
> Thank you for pointing this out. **The value 0.01 determines the final coefficient $\alpha$ at the last epoch and we expect the final value should be very close to 0.** If we use 0 instead, then the coefficient for the first epoch is not well defined since setting $s=0$ turns Equation 9 into $\alpha^t_\text{geometric}=\alpha_0(\frac{0}{\alpha_0})^0$.
>
> As you point out, we can indeed tune this value. Below we search this constant over the range $\{0.1, 0.05, 0.01, 0.005, 0.001\}$ with the experiment of Table 4 training ConvNeXt-Femto on STL-10. We can see that the best constant is 0.01 for all three initial coefficients. Therefore, we use 0.01 as a default choice. **To improve the presentation for Equation 9, we will replace the constant 0.01 with the symbol $\alpha_\text{final}$ in the revision.**
>
> |  | 0.1 | 0.05 | 0.01 | 0.005 | 0.001 |
> |---|---|---|---|---|---|
> | $\alpha_0=0.5$ | 1.628 | 1.627 |  **1.616** | 1.677 | 1.664 |
> | $\alpha_0=0.75$ | 1.601 |1.599 |  **1.582** | 1.629 | 1.631 |
> | $\alpha_0=1$ | 1.575 | 1.595 | **1.574** | 1.616 | 1.612 |
> &nbsp;
> ---
>
> We thank you for your valuable feedback and see resolving your questions and concerns as a great improvement to our paper. If you have any further questions or concerns, we are very happy to answer.

---

> > ### Author Response · Authors · 2024-11-28
> > **Kind reminder to review our rebuttal**
> >
> > Dear Reviewer Nknp,
> >
> > We hope this message finds you well. We are writing to kindly ask you to review our rebuttal. Please let us know if you have any further questions and we are happy to address them.
> >
> > Best,\
> > Authors

---

> ### Author Response · Authors · 2024-11-30
> **Kind reminder to review our rebuttal**
>
> Dear Reviewer Nknp,
>
> We hope this message finds you well. We kindly request your review of our rebuttal and would greatly appreciate your feedback. If you have any further questions or require additional clarification, please do not hesitate to let us know.
>
> Thank you for your time and consideration!
>
> Best regards,\
> Authors

---

### Official Review · Reviewer_QXir · 2024-11-06

**Soundness:** 4
**Presentation:** 4
**Contribution:** 3
**Rating:** 8
**Confidence:** 5

**Summary:**

This paper revisit SVRG in deep learning, showing how reasoning more carefully about control variates. The paper starts by a nice recap of what is believed to be variance reduction and SVRG performance in deep learning, and then guides the reader to the "additional parameter" solution. The authors complement their finding (i.e. a simple modification of SVRG can help training) experimentally on a diverse range of datasets in vision, using many architecture.

**Strengths:**

I like the paper, it is a simple idea yet presented very clearly:

1) The graphics and plots help the reader get the idea behind the paper. The many presented variance metrics also are very helpful in convincing the reader of the approach.

2) The authors go step by step, showing improved performance with "optimal variance reduction" and later showing why a simple coefficient can close most of the gap. This is very important, makes the investigation very scientific.

3) There are many experiments presented in the corresponding section.

**Weaknesses:**

I think this could be a very interesting easy reference for modern SVRG. I believe though some additional points can help delivering the idea in a more solid way:

1) As far as I understand, you are always using Adam with lr 3e-4 in your experiments. This is a quite nice lr - but it is well-known that Adam might profit from additional tuning. I do not believe your point here should be "you are better than Adam" - many papers overclaim performance as their main point, but you show much more. I think here what could be helpful is a plot of final accuracy or training loss over learning rare - comparing both with SGD and Adam. Showing this for 1 learning rate is not enough. Note that you do not have to show you are better than Adam, but just that you are better than SVRG and close most of the gap.

2) Figure 5 (SVRG with optimal alpha) is a bit weak: not much improvement over SGD id shown. Do you have an explanation for this? In a way this is the "best we can get" - yet in this specific example Adam, SGD and SVRG (with or withouth alpha) perform quite similar! the variance metrics also confirm that not much variance reduction is happening. I would personally try to find another example or compare this performance on multiple examples.

If the authors can show some improvements around these points, I am happy to raise my score!

-------

Authors addressed part of my concerns, and will likely improve experiments further, increased to accept.

**Questions:**

See above

---

> ### Author Response · Authors · 2024-11-23
> **Rebuttal by Authors**
>
> We sincerely thank you for your comment. We appreciate you find our work very scientific and can be a very interesting and easy reference for modern SVRG. We address your concerns and questions below:
>
> >W1: As far as I understand, you are always using Adam with lr 3e-4 in your experiments. This is a quite nice lr - but it is well-known that Adam might profit from additional tuning. I do not believe your point here should be "you are better than Adam" - many papers overclaim performance as their main point, but you show much more. I think here what could be helpful is a plot of final accuracy or training loss over learning rate - comparing both with SGD and Adam. Showing this for 1 learning rate is not enough. Note that you do not have to show you are better than Adam, but just that you are better than SVRG and close most of the gap.
>
> We wish to clarify that **the base learning rate for both the baseline (SGD/AdamW) and $\boldsymbol{\alpha}$-SVRG for all experiments in Section 5 is 4e-3**. This control setting ensures that any performance gain comes from $\alpha$-SVRG itself rather than any other factors (e.g., learning rate).
>
> Here we present results of $\alpha$-SVRG at different learning rates. Specifically, we sweep the learning rate for AdamW and $\alpha$-SVRG ($\alpha_0=0.5$) over the range of $\\{\text{1e-2}, \text{5e-3}, \text{1e-3}, \text{5e-4}, \text{1e-4}\\}$ and compare their final training loss [here](https://drive.google.com/file/d/1UABxiO6AUsMUwhUD6Tj_v591Qetu4_wr/view?usp=share_link) (click the link). **The results show $\alpha$-SVRG can decrease training loss better than vanilla AdamW in most cases.** We have incorporated this experiment in Appendix J.1 of the current draft.
>
> Note that in our experiments in Section 5, when comparing to the baseline (SGD / AdamW), $\alpha$-SVRG and standard SVRG use SGD / AdamW as the base optimizer. In other words, the variance reduced gradient from $\alpha$-SVRG is input into the update rule of the base optimizer (SGD / AdamW). Therefore, it is possible for $\alpha$-SVRG to perform better than AdamW since the variance reduction mechanism in $\alpha$-SVRG can help the base optimizer (AdamW) converge faster. **Nevertheless, we agree with you that the comparison to SVRG should be the main focus of the paper. In the revision, we will focus on the comparison to SVRG.**
>
> &nbsp;
> ---
> >W2: Figure 5 (SVRG with optimal alpha) is a bit weak: not much improvement over SGD id shown. Do you have an explanation for this? In a way this is the "best we can get" - yet in this specific example Adam, SGD and SVRG (with or without alpha) perform quite similar! the variance metrics also confirm that not much variance reduction is happening. I would personally try to find another example or compare this performance on multiple examples.
>
> We agree with you that the gap between SVRG with optimal coefficient and SGD baseline in original Figure 5 does not seem very large. This is because the scale of the training loss is very small when it has converged close to 0. **Here we provide better visualization of vanilla SGD, standard SVRG, SVRG with optimal coefficient, $\boldsymbol{\alpha}$-SVRG for Figure 5 in log scale [here](https://drive.google.com/file/d/1Ikb9Gyx4doRut_qi4Z_79E6NF8H6aR1C/view?usp=sharing).** We can see the gap is much larger than the original one. Qualitatively, SVRG with optimal coefficient achieves 17.7% lower final training loss (0.154) than SGD baseline (0.187). If we compare the final training loss of SVRG with optimal coefficient to standard SVRG (0.262), their gap is even larger with a 41.3% difference.
>
> **In terms of variance reduction, we would like to clarify that the variance reduction of SVRG with optimal coefficient (and $\boldsymbol{\alpha}$-SVRG) happens mostly during the early training (the first 5 epochs in Figure 5).** Quantitatively (see below), in Figure 5, the gradient variance of SVRG with optimal coefficient is about 40% lower than baseline at the beginning of the training.
>
> | | metric 1 | metric 2 | metric 3
> |---|---|---|---
> | SGD | 0.328 | 1.901 | 0.106
> SVRG (opt. coef.) | 0.217 | 1.104 | 0.061
> percentage drop | 33.7% | 41.9% | 42.5%
>
> **We show our above finding about the variance reduction of SVRG with optimal coefficient also applies to other datasets (e.g., Flowers and EuroSAT).** [Here](https://drive.google.com/file/d/1N1Q8UJuF3aNkpIUCUtRBHjJitT0M2sp3/view?usp=share_link) are the results. You can see on these datasets, the effects of gradient variance reduction and training loss decrease from $\alpha$-SVRG are more pronounced.
>
>
> &nbsp;
> ---
>
> We thank you for your valuable feedback and see resolving your questions and concerns as a great improvement to our paper. If you have any further questions or concerns, we are very happy to answer.

---

> > ### Comment · Reviewer_QXir · 2024-11-23
> > **Thanks!**
> >
> > Thanks so much for all efforts!
> >
> > I appreciate the efforts by the authors, I am a bit confused though about the following point:
> >
> > There is a bit of an improvement between alpha-SVRG and SVRG (the second plot you attached, which would greatly profit from multiple seeds): this "tiny bit" is enough to be better than Adam. I think placing a comparison with SGD and SVRG in the plots you attached (first link) would be of great interest!
> >
> > Note: "the base learning rate for both the baseline (SGD/AdamW) and alpha-SVRG for all experiments in Section 5 is 4e-3. This control setting ensures that any performance gain comes from alpha-SVRG" -- is very wrong. it is well known that optimal learning rate is hyperparameter-dependent: especially when comparing with Adam.
> >
> > Anyways, the paper is very well written and interesting. Please fix the above! I would recommend acceptance now, increasing my score.

---

> ### Author Response · Authors · 2024-11-23
> **Further Response by Authors**
>
> Thank you for your quick response to our rebuttal! We address your further concerns below:
>
> > Q1: There is a bit of an improvement between alpha-SVRG and SVRG (the second plot you attached, which would greatly profit from multiple seeds): this "tiny bit" is enough to be better than Adam. I think placing a comparison with SGD and SVRG in the plots you attached (first link) would be of great interest!
>
> We have just launched the experiments to compare SGD with $\alpha$-SVRG at different learning rates. We will update the results here once it is done. If it does not come out before the discussion deadline, we will still add them into our revision.
>
> &nbsp;
> ---
> > Q2: "the base learning rate for both the baseline (SGD/AdamW) and alpha-SVRG for all experiments in Section 5 is 4e-3. This control setting ensures that any performance gain comes from alpha-SVRG" -- is very wrong. it is well known that optimal learning rate is hyperparameter-dependent: especially when comparing with Adam.
>
> We agree with your point that learning rate is a very important hyper-parameter when comparing different optimizers. The optimal learning rate can be drastically different for $\alpha$-SVRG and SGD / AdamW. In response, we have included AdamW results in the Appendix J.1 and will incorporate the SGD results once the ongoing experiments are complete.
>
> &nbsp;
> ---
> We again thank you for your time and feedback on the paper! Your review has been very helpful in enhancing our paper.

---

> > ### Author Response · Authors · 2024-11-24
> > **Further Response by Authors**
> >
> > (Update results for Q1)
> > > Q1: There is a bit of an improvement between alpha-SVRG and SVRG (the second plot you attached, which would greatly profit from multiple seeds): this "tiny bit" is enough to be better than Adam. I think placing a comparison with SGD and SVRG in the plots you attached (first link) would be of great interest!
> >
> > We have updated the results to compare SGD and $\alpha$-SVRG at different learning rates [here](https://drive.google.com/file/d/1UdyavPxJiAAjxoqwRHUGdLYys8DZDWLb/view?usp=sharing). In most cases, $\alpha$-SVRG can decrease training loss better than SGD. We have added this comparison to Appendix J.1.
> >
> > &nbsp;
> > ---
> > We again thank you for your constructive comments! If you have any further concerns or questions, we are very happy to answer.

---

### Author Response · Authors · 2024-12-01
**Summary for Rebuttal**

We sincerely thank the reviewers for their insightful feedback and valuable suggestions, which have greatly contributed to the improvement of our submission during the rebuttal process. **We are encouraged that during the discussion period, three of the four reviewers have increased their ratings**: Reviewer QXir increases from 6 to 8, Reviewer 9C3X increases from 5 to 6, and Reviewer 4HSX increases from 3 to 5.

**We are also glad that Reviewer 4HSX (with an updated rating of 5 from 3) has acknowledged "most of the concerns were addressed, and empirical results demonstrated the potential of variance-reduced methods."** Also, in our most recent response, we tried to clarify the remaining questions regarding (i) the linear setup in Equation 2 and (ii) different schedulers to set the parameter $\alpha$ in the experiments.

In summary, our rebuttal includes the following:
- We demonstrate that using $\alpha$-SVRG only during early training achieves competitive performance with little additional computational costs. We also show that its performance is robust across different hyperparameter settings  (Reviewers 9C3X and 4HSX).
- We provide results of $\alpha$-SVRG with standard deviations across three seeds and show the statistical significance of its variance reduction and training loss decrease (Reviewers QXir and 4HSX).
- We verify our observations on $\alpha$-SVRG’s variance reduction and optimal coefficients held across different datasets (Reviewers QXir and Nknp).
- We conduct experiments showing the advantage of $\alpha$ over the baseline under different base learning rates and smaller batch sizes (Reviewers QXir and 4HSX).
- We show the feasibility of eliminating full batch gradient computation in $\alpha$-SVRG by using an EMA of past snapshot gradients (Reviewer 4HSX).
- We illustrate that $\alpha$-SVRG’s benefits are independent of other training strategies, such as learning rate warmup (Reviewer 4HSX).
- We clarify how to select the initial coefficient $\alpha_0$ (Reviewer Nknp).
- We add pseudocode for AdamW + $\alpha$-SVRG in the Appendix and correct several typos in the submission (Reviewers Nknp, 9C3X, and 4HSX).

We hope our responses can adequately address the reviewers' concerns. We will integrate all the work presented in the rebuttal into the revised submission, and we sincerely appreciate your valuable feedback.

---

### Meta-Review · Area_Chair_Hqd2 · 2024-12-19

**Metareview:**

The paper introduces α-SVRG, a variant of SVRG that integrates a coefficient to dynamically control the strength of variance reduction, decaying it over time. The reviewers appreciated the novelty of this approach and its demonstrated empirical effectiveness on various deep-learning tasks. The rebuttal and clarifications satisfied the reviewers. The consensus after discussion is that the empirical contributions and practical improvements are significant and novel enough to warrant acceptance.

**Additional Comments On Reviewer Discussion:**

During the reviewer discussion, the key points raised were:

1. Novelty of α-SVRG: Reviewers acknowledged the innovative idea of dynamically decaying the variance reduction strength.
2. Empirical Results: Concerns about the lack of theoretical justification were balanced by strong empirical evidence across deep-learning tasks.
3. Justification of Linear Decay: While questioned, the authors clarified its empirical motivation and effectiveness.

The rebuttal sufficiently addressed reviewer concerns, emphasizing practical contributions and performance gains. After weighing the novelty, empirical improvements, and discussions, the consensus favors acceptance.

---

### Decision · Program_Chairs · 2025-01-22

Accept (Poster)